# ControlAgent: Automating Control System Design via Novel Integration of LLM Agents and Domain Expertise

## Abstract

Control system design is a crucial aspect of modern engineering with far-reaching applications across diverse sectors, including aerospace, automotive systems, industrial processes, power grids, and robotics. Despite advances made by Large Language Models (LLMs) in various domains, their application in control system design remains limited due to the complexity and specificity of control theory. To bridge this gap, we introduce **ControlAgent**, a new paradigm that automates control system design via novel integration of LLM agents and control-oriented domain expertise. ControlAgent encodes expert control knowledge and emulates human iterative design processes by gradually tuning controller parameters to meet user-specified requirements for stability, performance (e.g. settling time), and robustness (e.g., phase margin). Specifically, ControlAgent integrates multiple collaborative LLM agents, including a central agent responsible for task distribution and task-specific agents dedicated to detailed controller design for various types of systems and requirements. In addition to LLM agents, ControlAgent employs a Python computation agent that performs complex control gain calculations and controller evaluations based on standard design information (e.g. crossover frequency, etc) provided by task-specified LLM agents. Combined with a history and feedback module, the task-specific LLM agents iteratively refine controller parameters based on real-time feedback from prior designs. Overall, ControlAgent mimics the design processes used by (human) practicing engineers, but removes all the human efforts and can be run in a fully automated way to give end-to-end solutions for control system design with user-specified requirements. To validate ControlAgent's effectiveness, we develop **ControlEval**, an evaluation dataset that comprises 500 control tasks with various specific design goals. Comparative evaluations between LLM-based and traditional human-involved toolbox-based baselines demonstrate that ControlAgent can effectively carry out control design tasks, marking a significant step towards fully automated control engineering solutions.

## 1 Introduction

Recent advancements in large language models (LLMs) have spurred the development of sophisticated LLM agents, demonstrating remarkable capabilities in areas such as code generation, reasoning, tool use, and software development, among many other applications (Hong et al., 2023; Zhang et al., 2024; Mei et al., 2024; Wu et al., 2023; Liu et al., 2023b; Talebirad & Nadiri, 2023; Li et al., 2023; M. Bran et al., 2024; Liu et al., 2024b; 2023a; Zhuge et al., 2024). Despite these breakthroughs, the application of LLM agents in modern engineering design remains relatively underexplored. Building on the exciting progress in LLM reasoning, it seems natural to expect great potential of LLMs as modern engineering design assistants. By breaking down complex engineering design processes into smaller specific tasks, LLM agents could potentially improve both the productivity and efficiency of engineering workflows via reducing human efforts from practicing engineers.

Control design is a cornerstone of modern engineering, underpinning a wide range of applications in both daily life and industrial processes, such as automobile cruise control systems, home thermostats, industrial robot manipulators, aircraft autopilots, chemical process control in refineries, and power grid frequency regulation (Åström & Murray, 2021; Ogata, 2009; Boyd & Barratt, 1991;

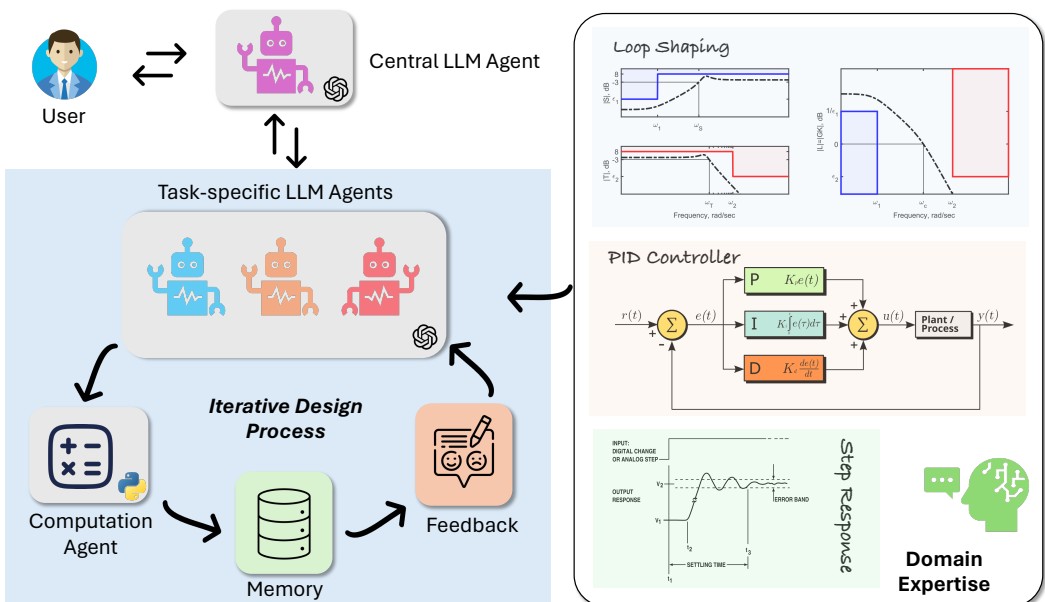

Figure 1: General **ControlAgent** framework.

Anderson, 1993; Rivera et al., 1986). Conventional controller design often requires human expertise and iterative design protocols, which may involve tedious repeated computation work. For instance, Proportional-Integral-Derivative (PID) control has been widely used in industry, but its design process involves iterative tuning from practicing control engineers to meet conflicting requirements[1] in terms of system performance and robustness (Ogata, 2009; Xu et al., 2008; Liu et al., 2014). It seems natural to ask whether LLMs can be leveraged to automate such tedious design processes and reduce the burden on human experts. In this paper, we provide an affirmative answer to this question via integrating LLM agents and control-oriented domain expertise in a novel manner.

Specifically, our paper presents **ControlAgent**, an LLM-based framework that automates control system design by seamlessly integrating domain knowledge and tool utilization. ControlAgent encodes expert control knowledge and emulates human iterative design processes by gradually tuning controller parameters to meet user-specified requirements for stability, performance (e.g. settling time), and robustness (e.g., phase margin). ControlAgent integrates multiple collaborative LLM agents, including a central agent for task distribution and task-specific agents for detailed controller design across various systems and requirements, alongside a Python computation agent that performs complex control gain calculations and evaluations based on standard design information provided by the task-specific LLM agents. Utilizing a history and feedback module, ControlAgent enables task-specific LLM agents to iteratively refine controller parameters, mimicking the design processes of practicing engineers while eliminating human effort to provide fully automated, end-to-end solutions for control system design that meet user-specified requirements. Figure 1 illustrates a general overview of the ControlAgent framework. Users simply provide the necessary task information, such as the dynamic systems to be controlled and the associated performance requirements. ControlAgent then analyzes the task, performs iterative design processes similar to practicing engineers, and returns the final design solution. Our contributions are threefold. Firstly, we present ControlAgent, a first fully automated LLM-based framework that emulates human-like iterative design processes for control engineering. By integrating domain-specific human expertise into LLM agents and combining external tool use, ControlAgent systematically refines control designs based on prior designs without human intervention. Secondly, we construct ControlEval, a thorough evaluation benchmark for classic control design, ranging from relatively simple first-order system designs to more complex higher-order system designs. This benchmark serves as a standard for evaluating LLM-based control design workflows. Thirdly, we conduct a comprehensive experimental study

---

[1]Due to the fundamental trade-offs between performance and robustness, control design is intrinsically subtle with a multi-objective nature. For example, classical control aims to achieve fast reference tracking and disturbance rejection while also being insensitive to noise and robust to model uncertainty.

on ControlEval to validate the performance and robustness of ControlAgent, demonstrating superior performance of ControlAgent over both LLM-based and traditional toolbox-based baseline methods.

**Unique Novelty.** Recently, there has been some work showing that LLMs have gained knowledge related to control engineering and can answer textbook-level control system questions to some extent (Kevian et al., 2024). However, going beyond the textbook level, LLMs still cannot generate practical control design in a reliable manner. Beside the computation errors, LLMs may also make various reasoning errors for practical control design. A key gap is that control design is intrinsically subtle due to the performance-robustness trade-off, and LLMs do not know how to mitigate such subtle trade-offs in a reliable way even if they are exposed to many different control methods. In this paper, we develop ControlAgent in a way that it mimics how practicing engineers mitigate such design trade-offs via PID tuning and frequency-domain loop-shaping (see Figure 1). Consequently, ControlAgent becomes reliable in designing controllers with satisfying performance and robustness.

## 2 RELATED WORK

**Classic Control Design.** Controller design is traditionally approached in a case-by-case manner, as it heavily depends on the specific applications at hand. Among various control strategies, PID control and loop-shaping remain the most widely used due to their simplicity and ease of implementation. Over the years, a plethora of PID/loop-shaping tuning methods have been developed (Åström & Hägglund, 1995; Skogestad, 2001; Mann et al., 2001; Awouda & Mamat, 2010; O'dwyer, 2009; Padula & Visioli, 2011; Panda, 2008; Lequin et al., 2003; Skogestad, 2003). Despite these advancements, the tuning process still heavily relies on human expertise and manual intervention to identify suitable controller parameters that meet design criteria. ControlAgent aims to fill this gap via integrating LLM agents and human expert knowledge for automating control system design.

**LLM for Engineering Design.** Several studies have explored the potential of LLMs in addressing various engineering domains (Ghosh & Team, 2024; Poddar et al., 2024; Alsaqer et al., 2024; Majumder et al., 2024). In addition, (Kevian et al., 2024; Syed et al., 2024; Xu et al., 2024) introduced benchmark datasets to evaluate the textbook-level knowledge of LLMs in control, transportation, and water engineering. AnalogCoder (Lai et al., 2024) is developed for analog circuits design, while SPICED (Chaudhuri et al., 2024) focused on the bug detection in circuit netlists with the aid of LLMs. Furthermore, AmpAgent (Liu et al., 2024a) utilizes LLMs for multi-stage amplifier design.

**LLM-based Agents.** LLM-based agents take textual or visual information as input for complex task solving, which has attracted a lot interests in both academia and industry recently (Wang et al., 2024). In particular, multi-agent systems leverage the interaction among multiple LLM agents for more complex tasks (Kambhampati et al., 2024; Zhuge et al., 2024; Josifoski et al., 2023; Park et al., 2023; Li et al., 2023; Zhuge et al., 2023). For example, AutoGen (Wu et al., 2023) provides a generic multi-agent framework for various applications including coding, question answering, mathematics, etc. MetaGPT (Hong et al., 2023) is a multi-agent LLM framework inspired by the Standardized Operating Procedures developed from human protocol for efficient task decomposition and coordination. Overall, the field of LLM agents is very active. See Appendix B.1 for a more comprehensive literature review.

## 3 PRELIMINARY

This section briefly reviews the basic background of classic control. The field of control engineering focuses on the design, analysis, and implementation of feedback mechanisms that are used to regulate and steer dynamic systems to achieve desired outputs or behaviors (Åström & Murray, 2021; Ogata, 2009). Application examples includes everyday devices like the heating and air conditioning as well as more advanced systems such as autonomous cars and airplane autopilots. First, we review the notion of dynamical systems studied in classic control. A dynamical system can be represented in various forms including differential equations, state-space models, and transfer functions (Goodwin et al., 2001; Boyd & Barratt, 1991). The main objects studied in classic control design are linear time-invariant (LTI) systems, which can be represented in either time domain by a linear ordinary differential equation (ODE) or in frequency domain by an equivalent transfer function. For instance, the transfer function of an LTI system has the following form:

$$G(s) = \frac{b_m s^m + b_{m-1} s^{m-1} + \cdots + b_1 s + b_0}{a_n s^n + a_{n-1} s^{n-1} + \cdots + a_1 s + a_0}, \tag{1}$$

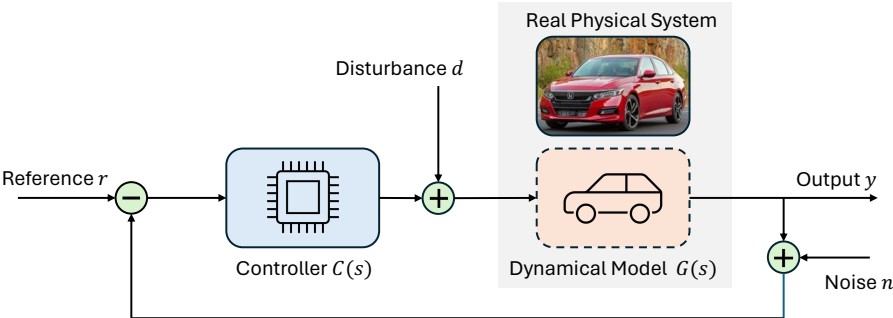

Figure 2: A feedback control system illustrating the reference $r$, measured output $y$, disturbance $d$, and noise $n$. The dynamical model $G(s)$ provides a mathematical approximation of the real physical system. The inherent mismatch between the real system and its mathematical model underscores the need for a robust controller $C(s)$ to ensure reliable performance despite modeling inaccuracies.

where $s$ is the complex frequency variable in the Laplace domain. Notice that the system (1) has an equivalent time-domain ODE form that relates the input signal $u(t)$ to the output signal $y(t)$:

$$a_n \frac{d^n}{dt^n} y(t) + \ldots + a_1 \frac{d}{dt} y(t) + a_0 y(t) = b_m \frac{d^m}{dt^m} u(t) + \ldots + b_1 \frac{d}{dt} u(t) + b_0 u(t) \qquad (2)$$

This form is general enough to model the dynamics of various practical systems such as automotive systems, robotics, and many others. There always exist gaps between models and reality. Classic control is successful in practice as control engineers use robustness margins to account for such gaps.

**Classic Control Design.** Feedback control, shown in Figure 2, can be used to steer the plant output $y(t)$ to track a reference signal $r(t)$. This architecture: (a) uses a sensor to measure the output $y(t)$, (b) computes the tracking error $e(t) = r(t) - y(t)$, and (c) uses a control algorithm $C(s)$ to compute the input to the plant based on the error. Figure 2 depicts a standard feedback loop where the measured output $y(t)$ is used by the controller to compute the input to the system which then affects the output $y(t)$. Classical control focuses on designing the controller $C(s)$ (which is an LTI system by itself). There are numerous, often conflicting, objectives for classic control design, and standard design requirements include[2]: i) closed-loop stability, ii) fast reference tracking, iii) rejection of disturbance (e.g., the wind gusts and hills acting on a car), iv) actuator limits, v) rejection of sensor noise, and vi) robustness to model-reality gap (e.g., unmodeled dynamics, etc). This necessitates the performance/robustness trade-offs, which lie at the core of classic control design.

**Performance/Robustness Trade-offs.** The various design requirements roughly boil down to three main categories : i) closed-loop stability , ii) performance (e.g. tracking speed), and iii) robustness (see Appendix B.2 for definitions). The **settling time** $T_s$ is arguably the most important performance metric, since it measures the time required for the systems response to reach within a specified percentage (e.g., 2% or 5%) of the steady-state value, and small $T_s$ just implies fast reference tracking. Robustness is also crucial. As illustrated in Figure 2, there always exist a gap between the dynamical model used for control design and the real physical system which the controller is deployed on. It is a must to make the controller robust against the model-reality gap. In this context, **phase margin** is typically recognized as the most important robustness metric (Chang & Han, 1990; Ho et al., 1996), and on the conceptual level, large phase margin implies strong robustness. Since achieving small settling time (fast tracking) and large phase margin (robustness) are competing objectives, practicing control engineers typically use settling time and phase margin as tuning knobs for navigating the performance/robustness trade-offs in classic control design. There are many secondary metrics (e.g. gain margin, etc) that can be used to provide fine-grained descriptions for the six control design requirements mentioned above. However, it is known from control practice that PID and loop-shaping design with settling time and phase margin being tuning knobs can be sufficient in addressing the complicated performance/robustness trade-offs involving all six control design requirements simultaneously. **One main goal of our study is to develop the first LLM-based framework that can automatically address such subtle performance/robustness trade-offs in control design.**

---

[2]See Appendix B.2 for more explanations of these control design requirements.

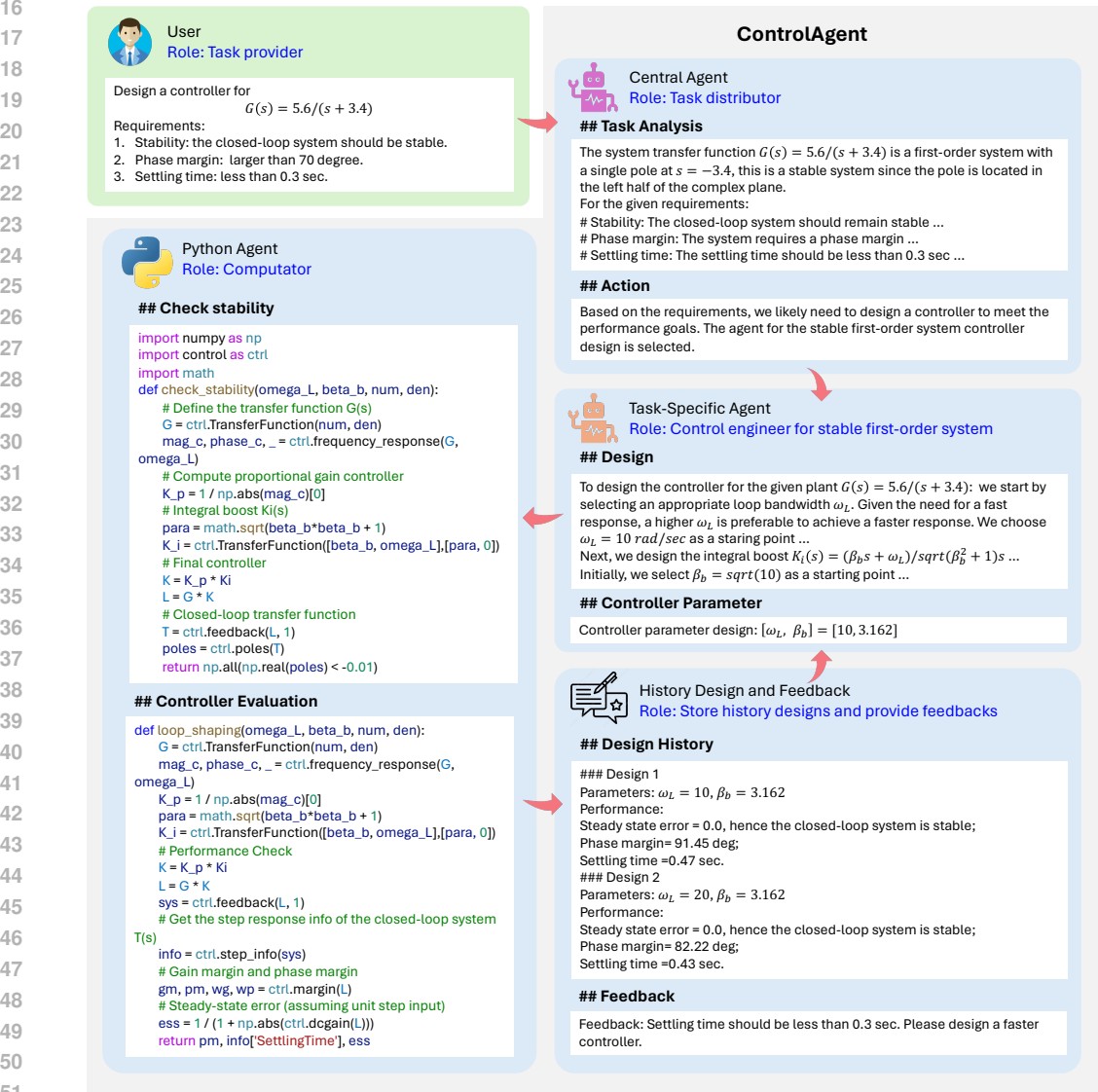

Figure 3: The controller design process of ControlAgent, showcasing interactions between the User, Central agent, Python agent, History and Feedback module, and Task-Specific Agents to design a controller that meets stability, phase margin, and settling time requirements.

# 4 CONTROLAGENT

In this section, we present ControlAgent, detailing its agent architecture, iterative design mechanisms, and communication protocols. An overview of ControlAgent has been illustrated in Figure 1,

**Agent Design.** We break down the complex controller design into smaller and more specific tasks, requiring the collaboration of agents with different skills and expertise. ControlAgent compromises three types of agents: 1) **Central agent** $\mathcal{A}_c$ acts as the task distributor, processes user inputs and assigns specific requests to the sublevel agents based on the nature of the controller design task, 2) **Task-specific agent** $\mathcal{A}_{spec}$ receives the user request and high-level task analysis from the central agent, and encodes with domain-specific expertise to initiate the controller design process, following the iterative methodology discussed below, and 3) **Python computation agent** $\mathcal{A}_p$ carries out the complex computation steps involved in controller design and performance evaluations, ensuring reliable controller synthesis and evaluations. Figure 3 present an illustrative example of the controller design workflow within ControlAgent. The user initially provides the system's dynamic

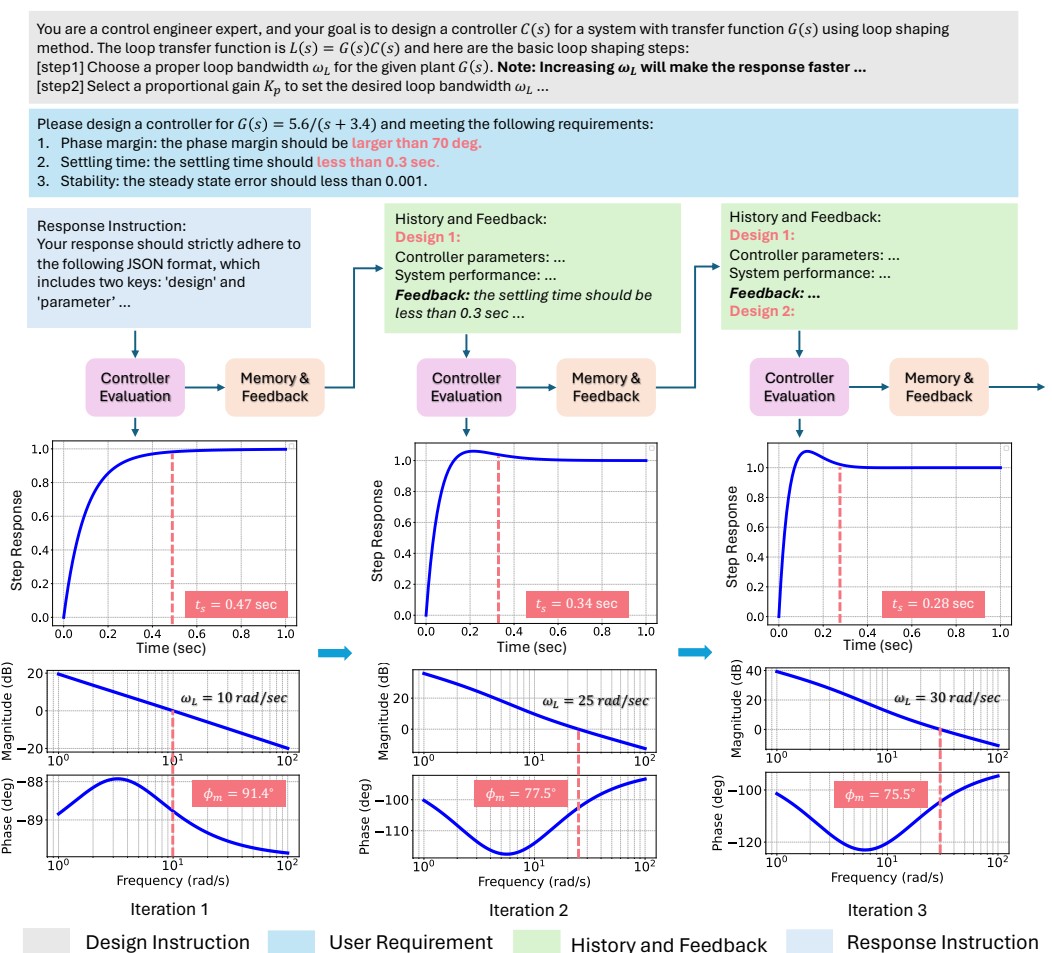

Figure 4: Workflow of the task-specific agent in ControlAgent. The design history and feedback are dynamically updated based on previous iterations. ControlAgent refines its designs iteratively, incorporating user instructions and feedback at each step. By the third iteration, ControlAgent achieves a final design that satisfies the users requirements, achieving a settling time of less than 0.3 seconds (as shown in the time response plot) and maintaining a phase margin consistently greater than 70° (as depicted in the Bode plot).

model (represented as a transfer function) along with the specified design criteria on closed-loop stability, settling time, and phase margin. The central agent subsequently analyzes the task and delegates it to a specialized task-specific agent, tailored to the task's requirements. Each task-specific agent, endowed with domain-specific expertise, initiates the design process upon receiving the assignment. The designed controller is evaluated by the Python agent, while a history and feedback module archives the design process and generates valuable feedback to enable iterative refinement.

**Iterative Design via Structured Memory Design.** ControlAgent relies on the iterative design and feedback mechanism to mimic the design processes used by practicing engineers (see Figure 4). Traditional controller design by control engineers often involves a cycle of trial and error, requiring fine-tuning of controller parameters based on observed feedback. Similarly, for LLM agents to perform control system design effectively, they must follow an iterative design process. This involves accessing previous designs and performance metrics, and using feedback to refine their outputs to improve the performance and robustness of the controller configuration. However, storing all past outputs of LLM agents and simply reusing them in the next iteration is impractical due to the context window limitations of LLMs. To address this, ControlAgent manages memory through an efficient structured memory buffer $\mathcal{M}$ that retains only essential information: the previously designed controller parameters and their associated performances, rather than complete historical outputs. This

| System Type | 1st-order stb | 2nd-order stb | 1st-order w/ delay | Higher-order System |
|---|---|---|---|---|
| System Model | $\dfrac{2.19}{s + 10.99}$ | $\dfrac{5.88}{s^2 + 1.43s + 0.91}$ | $\dfrac{8.79e^{-0.14s}}{s + 4}$ | $\dfrac{225}{s^3 + 14.2s^2 + 46s + 40}$ |
| Response Mode | Moderate | Slow | (-) | (-) |
| Stability | $\checkmark$ | $\checkmark$ | $\checkmark$ | $\checkmark$ |
| Settling Time Range | $T_s \in [0.04, 0.58]$ | $T_s \in [12.70, 34.04]$ | $T_s \in [0.63, 6.68]$ | $T_s \in [1.05, 8.4]$ |
| Phase Margin | $\phi_m \geq 81.74°$ | $\phi_m \geq 61.57°$ | $\phi_m \geq 44.06°$ | $\phi_m \geq 62.54°$ |

Table 1: System models and their corresponding control design criteria.

strategy allows the agent to recall crucial details from past iterations without exceeding memory capacity. In addition, ControlAgent also dynamically evaluates the current performance in comparison to user requirements. If the current design does not meet the requirements, a feedback $\mathcal{F}$ is created, encoded, and then incorporated into the input prompt for the LLM agent in the next iteration.

Now we explain Figure 4, which illustrates the iterative design process. The input prompt to the LLM agents consists of four main components: 1. **Design instruction**: the design instruction $\mathcal{E}_{\text{spec}}$ is distilled from domain expertise for each specific task to enhance the LLM agents' capabilities in controller design with particular focus on mitigating performance/robustness trade-offs via PID or loop-shaping with settling time and phase margin being used as the tuning knobs. 2. **User requirements**: the user requirements $\mathcal{U}$ are provided directly by the user. 3. **Memory and feedback**: this component includes the retrieval of previous design parameters from the structured memory buffer $\mathcal{M}$, along with automatically generated feedback to highlight the deficiencies of the current design. 4. **Response instruction**: the response instruction $\mathcal{R}$ specifies the response format to ensure that key information can be extracted efficiently. Upon receiving the task requirements from the central agent $\mathcal{A}_c$, the task-specific LLM agent $\mathcal{A}_{\text{spec}}$ iteratively designs a new controller based on the provided instructions, previously failed designs, and feedback. During each iteration, $\mathcal{A}_{\text{spec}}$ generates a new controller design, which is then stored in the memory buffer. Subsequently, a Python agent $\mathcal{A}_p$ retrieves the design and conducts evaluations. If the current design satisfies the user-defined requirements, the iteration process halts, and the successfully designed controller is returned. Otherwise, a feedback signal is generated by comparing the current performance against the user requirements, and the process continues to the next iteration until the maximum iteration count is reached. The iterative design process of ControlAgent is summarized in Algorithm 1 at Appendix.

## 5 CONTROLEVAL

Since no suitable open-source dataset is currently available for validating ControlAgent, we developed a new evaluation dataset, called **ControlEval**, to serve this purpose. ControlEval consists of 10 distinct types of control tasks based on various systems and requirements. For each task type, we construct 50 individual systems, each paired with its corresponding design requirements, resulting in a comprehensive dataset of 500 control tasks. ControlEval includes a diverse set of dynamical systems such as first-order stable and unstable systems, second-order stable and unstable systems with varying response speed modes, first-order systems with time delay, and general higher-order systems. The design criteria for each task involve a combination of closed-loop stability, settling time (to quantify tracking performance), and phase margin (to assess robustness). These are three key metrics for classic control design. For first and second-order stable systems, we further differentiate between three different speeds of response defined by the variation in settling time: *fast*, *moderate*, and *slow*. The fast mode requires the system to converge to its steady-state value within a short period of time, which is typical for applications that demand quick response times, such as servo motor control systems (Krah & Klarenbach, 2010) and quadcopter flight control systems (Bramlette & Barrett-Gonzalez, 2017). In contrast, the slow mode requires a more gradual convergence, which is more suitable in scenarios where the dynamic system model is less precise and less aggressive control is desired, such as wind turbine control (Ossmann et al., 2021). Some samples from ControlEval are provided in Table 1 including the system types, system dynamical models, response mode, and the associated design requirements.

Due to inherent limitations in the control of unstable systems, systems with time delays, and higher-order systems, it is not always possible to satisfy arbitrary combinations of performance/robustness requirements (Stein, 2003; Seron et al., 2012; Freudenberg & Looze, 1985; 1987; 1988). Therefore,

| System Type | 1st-ord stb | | | 2nd-ord stb | | | 1st-ord ustb | 2nd-ord ustb | w/ dly | Hgr-ord |
|---|---|---|---|---|---|---|---|---|---|---|
| Response Mode | fast | moderate | slow | fast | moderate | slow | (-) | (-) | (-) | (-) |
| Zero-shot | 8.0 | 19.2 | 10.0 | 14.0 | 18.4 | 13.2 | 5.2 | 0.4 | 15.6 | 2.0 |
| Zero-shot CoT | 26.8 | 3.2 | 0.4 | 12.4 | 18.8 | 12.0 | 4.4 | 0.8 | 8.8 | 8.0 |
| Few-shot | 12.4 | 19.6 | 15.6 | 12.0 | 12.4 | 15.2 | 14.0 | 29.2 | 11.6 | 12.0 |
| Few-shot CoT | 11.2 | 21.6 | 21.2 | 7.6 | 14.0 | 25.6 | 6.0 | 22.4 | 16.0 | 16.4 |
| PIDtune | 56.0 | 90.4 | 86.4 | 81.6 | 98.8 | 77.6 | 30.4 | 10.8 | **100.0** | 50.0 |
| ControlAgent | **100.0** | **100.0** | **100.0** | **100.0** | 98.8 | **90.8** | **97.2** | **96.8** | 97.2 | **82.0** |

Table 2: Average Success Rate (ASR, %) of baseline methods and ControlAgent on ControlEval for various system types and response modes. The best result for each task is highlighted in bold. The results show that ControlAgent consistently outperforms all other LLM-based and toolbox-based baselines (except the first-order system with delay) across all categories, demonstrating its effectiveness and robustness in handling diverse control tasks.

human experts have carefully curated the dataset to ensure that the task requirements are feasible and achievable. Further information on the dataset can be found in Appendix F.

# 6 EXPERIMENTAL RESULTS

In this section, we present a comprehensive set of experiments to evaluate the performance of ControlAgent on the ControlEval. GPT-4o is used as the main underlying base LLM for both the central agent and task-specific agents, and study on comparing different base LLMs is also presented. The detailed prompts for ControlAgent can be found in Appendix E.1. Additionally, we compare ControlAgent against two different baseline categories: LLM-based and control toolbox-based baselines.

**LLM-based Baselines:** We consider four LLM-based baseline approaches utilizing GPT-4o: zero-shot prompting, zero-shot Chain-of-Thought (CoT), few-shot, and few-shot CoT. In the zero-shot approach, we directly provide the user requirements and ask the LLM to perform the controller design without additional guidance. The CoT variant enhances this by prompting the LLM to explicitly conduct the design step-by-step. For the few-shot approach, we present the LLM with several examples of successful controller designs to guide its process. In the few-shot CoT setting, the prompt not only includes the successful designs but also details the step-by-step reasoning process required to create a successful controller. The detailed prompt for each setting can be found in Appendix E.2.

**Control Toolbox-based Baseline:** We also considered the widely used control toolbox for PID design: PIDtune (MathWorks, 2023) from MathWorks as a baseline. This toolbox is human-involved as the user needs to specify a proper value of crossover frequency as an input to optimize the controller gains, whereas ControlAgent tunes crossover frequency automatically without any human effort. Further details on how we set up PIDtune are reported in Appendix D.

**Evaluation Metrics:** We use **Average Successful Rate (ASR)** to measure the effectiveness of control designs across multiple independent trials for each method, and we use **Aggregate Successful Rate (AgSR)** to evaluate the success designs on a system-by-system basis, where one system is considered successfully designed if at least one of the multiple independent trials results in a successful controller design. We also employed the standard $pass@k$ with $k = \{1, 3, 5\}$ to provide a more robust metric with reduced variance. The formal metric definitions can be found in Appendix D.

## 6.1 MAIN RESULTS

Table 2 shows the ASR of ControlAgent and various baseline methods on the ControlEval benchmark. The best results for each task are highlighted in bold. Our key observations are given below.

**ControlAgent consistently outperforms all baseline methods.** ControlAgent achieves significantly higher ASR across all control tasks compared to both LLM-based and traditional toolbox-based baselines (with the sole exception of the first-order system with time delay, where ControlAgent achieves the second-best result at 97.2%). This superior performance is evident not only for simpler first-order and second-order stable systems but also for more complex cases, such as unstable systems and higher-order systems. The ability of ControlAgent to maintain high success rates across diverse system types showcases the potential of integrating LLMs with domain expertise, making it a highly reliable tool for automated control system design.

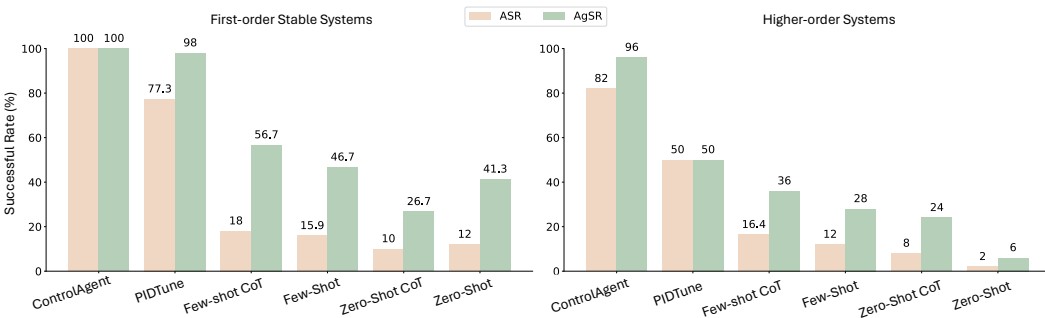

Figure 5: ASR and AgSR for first-order stable systems (averaged across fast, moderate, and slow modes) and higher-order system.

|  | ControlAgent | | w/o-iterative | | w/o-instruction | | w/o-python agent | | w/o-feedback | |
|---|---|---|---|---|---|---|---|---|---|---|
|  | ASR | iteration # | ASR | iteration # | ASR | iteration # | ASR | iteration # | ASR | iteration # |
| fast | **100.0** | **2.74** | 28.4 | (-) | 70.4 | 4.84 | 76.0 | 4.34 | 85.6 | 4.45 |
| moderate | **100.0** | **1.78** | 33.2 | (-) | 60.4 | 6.38 | 85.2 | 3.71 | 92.4 | 3.05 |
| slow | **100.0** | **2.19** | 4.0 | (-) | 56.4 | 6.39 | 71.2 | 5.19 | 94.0 | 2.46 |

Table 3: Ablation study results (ASR and average iteration number) for ControlAgent and its various component configurations. The ablated versions exclude specific components, such as iterative refinement, user instructions, the Python agent, and feedback incorporation.

**ControlAgent can solve easy tasks perfectly.** For relatively simpler systems, such as first/second-order stable systems, ControlAgent achieves perfect scores (100% ASR) across all response modes (fast, moderate, and slow). This indicates that ControlAgent is capable of flawlessly handling straightforward control problems, meeting all user-defined performance requirements.

**PIDtune outperforms LLM-based baselines on most control tasks.** It is noteworthy that PIDtune, a control toolbox-based method, performs better than LLM-based baselines (e.g., Zero-shot and Few-shot) on most control tasks, except for second-order unstable systems. This suggests that LLMs alone or simple prompt engineering methods are not sufficient to solve many control tasks effectively. The results highlight the gap between standard LLM capabilities and traditional control toolboxes. ControlAgent bridges this gap by employing an iterative controller design procedure that integrates LLMs, control domain expertise, and tool utilization to mimic how practicing control engineers mitigate the performance/robustness trade-offs in classic control design.

Figure 5 illustrates the ASR and AgSR for ControlAgent and the baseline methods for first-order stable systems (averaged across fast, moderate, and slow modes) and higher-order system, respectively. We run each method for five independent trials. The results show that each method significantly improves its success rate, highlighting the advantage of aggregating results from multiple trials to boost overall performance. ControlAgent remains one of the top-performing methods, achieving high success rates across all methods. More AgSR results can be found in Appendix D.

## 6.2 ABLATION STUDY

In this section, we perform ablation study on the ControlAgent.

**Effect of Key Components in ControlAgent:** To investigate the impact of different components within ControlAgent on its overall performance, we compare the ASR and the average number of iterations required for successful design across three response modes for first-order stable systems. The results, shown in Table 3, indicate that the complete version of ControlAgent achieves a perfect ASR (100%) across all response modes with the fewest iterations, underscoring the effectiveness of its integrated design. In contrast, removing the iterative design process leads to a drastic decline in ASR, particularly for the slow response mode, where the ASR drops to just 4%. Similarly, excluding design instructions or the Python agent significantly reduces the ASR and increases the number of iterations needed for success, highlighting the critical role these components play in improving design efficiency. Although ControlAgent performs reasonably well without feedback, the increased

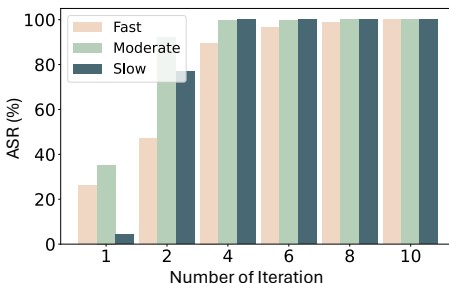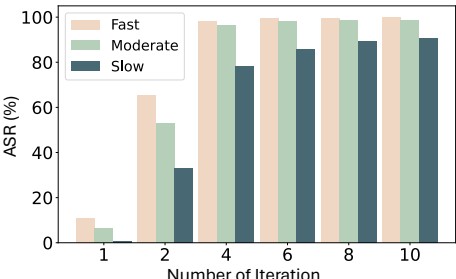

Figure 6: The effect of the number of iterations on ASR across different response modes (Fast, Moderate, and Slow). Left: first order stable systems; right: second order stable systems.

| Base LLM | GPT-4o | | Claude-3.5 Sonnet | | GPT-4-turbo | | Gemini-1.5-pro | | GPT-3.5-turbo | |
|---|---|---|---|---|---|---|---|---|---|---|
| | ASR | iteration # | ASR | iteration # | ASR | iteration # | ASR | iteration # | ASR | iteration # |
| **fast** | **100.0** | 2.74 | 98.4 | **2.66** | 94.0 | 3.82 | 86.8 | 2.96 | 49.2 | 6.84 |
| **moderate** | **100.0** | **1.78** | 99.2 | 2.05 | 98.4 | 2.55 | 86.4 | 2.41 | 97.2 | 3.01 |
| **slow** | **100.0** | 2.19 | 97.2 | **2.18** | 99.2 | 2.14 | 85.6 | 2.67 | 77.6 | 4.18 |

Table 4: ASR (%) and average number of iterations for ControlAgent using different base LLMs across three response modes (fast, moderate, and slow) for first-order stable systems. The highest ASR and lowest iterations highlighted in bold.

average iteration count shows that feedback is essential for faster convergence. Overall, these findings demonstrate that each component is vital to the robustness and efficiency of ControlAgent. Figure 6 demonstrates that increasing the maximum number of iterations consistently improvement in ASR across all response modes (fast, moderate, and slow) for first-order and second-order stable systems. As the number of iterations increases, ControlAgent has more opportunities to refine its design, which translates into higher success rates. This trend indicates that allowing more iterations enhances ControlAgent's ability to meet control design criteria, particularly for complex scenarios that may require additional iterations to achieve optimal results.

**Results on Different Base LLMs:** Table 4 presents the performance of ControlAgent with different base LLMs, including GPT-4o, Claude-3.5 Sonnet, GPT-4-turbo, Gemini-1.5-pro, and GPT-3.5-turbo. The results indicate that all state-of-the-art LLMs achieve reasonably good performance, with most models attaining high ASR values across different response modes. GPT-4o stands out by achieving a perfect ASR (100%) in all response modes and requiring the fewest iterations in the moderate mode. Similarly, Claude-3.5 Sonnet and GPT-4-turbo perform competitively; notably, Claude-3.5 achieves near-perfect ASR and has the lowest iteration count for the fast and slow modes. Although there is still a performance gap for Gemini-1.5-pro and GPT-3.5-turbo, these findings suggest that the state-of-the-art LLMs perform similarly, demonstrating that ControlAgent is flexible and adaptable to a variety of LLM configurations.

## 7 LIMITATIONS AND FUTURE WORK

In this paper, we introduced ControlAgent, an advanced LLM-powered framework for automated control system design. Despite the strong performance of ControlAgent across a range of control tasks, several limitations indicate avenues for future research and enhancement. One primary constraint is that the current implementation of ControlAgent is tailored to LTI systems and conventional control strategies, such as loop-shaping and PID controllers. Future work can expand ControlAgents capabilities by considering complex nonlinear systems and integrating advanced control strategies, such as adaptive and robust controllers. Another compelling direction involves utilizing different base LLMs for distinct roles, leveraging their unique strengths and expertise. For instance, incorporating fine-tuned, smaller LLMs for specialized tasks within control system design could improve efficiency and reduce dependence on proprietary models. Finally, the evaluation dataset, ControlEval, could be further extended to include more complex control tasks, such as real-world systems and hardware implementations, providing a more comprehensive assessment of ControlAgent's practical utility. We provide more detailed discussions on the future research directions in Appendix A.

ETHICS STATEMENT

In developing ControlAgent, we carefully considered the ethical implications of our work and took steps to ensure responsible research practices. All experiments were conducted using simulation environments and synthetic datasets, with no involvement of human subjects, thereby avoiding any privacy, security, or legal compliance concerns. ControlAgents focus on automated control system design raises the possibility of its deployment in critical applications, such as industrial automation, autonomous vehicles, and robotics. Improper use or deployment of AI-driven control systems in such domains could result in unintended outcomes. To mitigate these risks, we emphasize the need for rigorous testing, validation, and adherence to established safety standards before applying ControlAgent to real-world systems.

In terms of transparency and accessibility, the use of proprietary LLMs may limit broader access and reproducibility. To address this, future work will explore the use of open-source LLMs to enhance accessibility and facilitate community collaboration. Additionally, since ControlAgents performance depends on LLMs that could inherit biases from their training data, ongoing research will focus on mitigating bias and ensuring fairness in control system recommendations. The authors declare no conflicts of interest or external sponsorship that influenced the findings or interpretations in this study. Overall, we are committed to developing and applying ControlAgent ethically, with careful consideration of its societal impact and potential risks.

REPRODUCIBILITY STATEMENT

We have taken several measures to ensure the reproducibility of our results. Detailed descriptions of the experimental setup, hyperparameters, and configurations are provided in Section 6 of the main paper and Appendix D. Specifically, the architecture of ControlAgent and the LLM prompt structures are outlined in Section 4 and Appendix E. We also provide comprehensive descriptions of the ablation studies and baseline comparisons in Section 6 to aid in reproducing the results. The dataset used in our experiments, including details on dataset generation and control design criteria, is thoroughly described in Appendix F. Additionally, our code is available through an anonymized link for reproducibility check: https://anonymous.4open.science/r/ControlAgent-C5A1/.

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

## A  CONTROLAGENT: FUTURE OUTLOOK

In this section, we explore the future prospects of ControlAgent. We believe that ControlAgent represents a foundational initial step toward automated control system design using LLMs. Further research is necessary to expand its capabilities, enabling it to tackle more complex and realistic control challenges.

### A.1  EXPANSION TO NONLINEAR SYSTEMS AND ADVANCED CONTROL STRATEGIES

The current scope of ControlAgent is limited to Linear Time-Invariant (LTI) systems and conventional control strategies, which, although widely used in many industrial applications, restrict its applicability to a subset of control problems. However, in real-world scenarios, many systems exhibit nonlinear behavior, time-varying dynamics, or other complexities that are not sufficiently captured by LTI models. Future research should aim to incorporate advanced control strategies, such as nonlinear control methods (Sastry, 2013) (e.g., Lyapunov control, sliding mode control, backstepping, etc.), as well as adaptive and robust control frameworks (Zhou & Doyle, 1998). Expanding ControlAgent to handle these complex dynamics would significantly broaden its applicability to industries requiring sophisticated control solutions, such as robotics, aerospace, and automotive engineering. Additionally, leveraging the creative potential of LLMs could lead to innovative control strategies beyond the scope of traditional human-designed approaches (Tian et al.; Gómez-Rodríguez & Williams, 2023).

### A.2  MODULAR INTEGRATION OF DIFFERENT LLMs

The architecture of ControlAgent currently relies on a single base LLM for both central LLM agent and task-specific LLM agent. A promising research direction involves the modular integration of various LLMs based on their specific expertise. For example, specialized LLMs fine-tuned for mathematical reasoning, optimization, or control theory could be assigned to different roles within the overall framework of ControlAgent. This modular approach could leverage smaller, more focused models to handle niche aspects of control design. In addition, using open-source LLMs for non-critical tasks would reduce the reliance on proprietary models, making ControlAgent more accessible and adaptable.

### A.3  EXTENDING THE CONTROLEVAL DATASET FOR COMPREHENSIVE VALIDATION

ControlEval includes various control tasks that predominantly feature LTI systems. Extending ControlEval to include more complex tasks, such as real-world control systems and hardware-in-the-loop simulations, would provide a more robust validation of ControlAgents capabilities. Additionally, including scenarios that test the robustness and adaptability of ControlAgent to external disturbances, model uncertainties, and unmodeled dynamics would further establish its practical utility and readiness for real-world deployment.

## B  MORE DISCUSSIONS ON RELATED WORK AND CONTROL BACKGROUND

### B.1  MORE RELATED WORK

**Classic Control Design**: PID controllers have been a cornerstone of control system design. The widespread adoption of PID controllers is attributed to their simplicity, robustness, and effectiveness in managing a wide range of dynamic systems. Theoretical advancements have focused on optimizing PID parameters to achieve a desired performance, with methods such as Ziegler-Nichols tuning rules (Ziegler & Nichols, 1942) providing a heuristic-based starting point for controller tuning. Over the years, a range of adaptive and robust PID tuning techniques have been proposed, extending the PID controller's applicability to nonlinear, time-varying, and uncertain systems (Ang et al., 2005; Åström & Murray, 2021).

Loop shaping is another powerful approach to control system design, rooted in frequency domain techniques and aimed at shaping the open-loop transfer function to achieve specific performance and robustness goals (Ogata, 2009). The central idea behind loop shaping is to design controllers

that provide sufficient bandwidth, disturbance rejection, and stability margins by directly manipulating the system's frequency response. Loop shaping approaches use tools like Bode plots to tailor the system's gain and phase characteristics (Doyle et al., 2013). The importance of loop shaping is evident in its continued application across various industrial domains, including process control (Morari & Zafiriou, 1989), aerospace (Blight et al., 1994), and mechatronics (Ohnishi, 1996), showcasing its effectiveness in addressing real-world control challenges. Nevertheless, all the existing control design methods still heavily rely on the domain expertise and human intuition. ControlAgent makes an meaningful initial step towards automating the control system design by integrating LLM agents and human expert knowledge.

**LLMs for Engineering Design:** LLMs are increasingly being explored across various engineering domains due to their versatility and capacity for solving complex tasks. In the domain of electric grids, for instance, GridFM (Hamann et al., 2024) has been introduced as a foundation model capable of addressing a wide range of challenges, such as power flow estimation, grid expansion planning, and electricity price forecasting. Similarly, an agent-based framework proposed in (Jia et al., 2024) leverages techniques such as Chain-of-Thought (CoT) and Retrieval-Augmented Generation (RAG) to enhance LLMs' ability to perform power system simulations using previously unseen tools. In software engineering, LLM4SE (Hou et al., 2023) provides a comprehensive survey on the application of LLMs in this domain, showcasing their achievements so far while also identifying open challenges and promising future research directions. For materials science, models like MatBERT (Trewartha et al., 2022), a variant of the BERT architecture, and MatSciBERT (Gupta et al., 2022), trained on a vast corpus of materials science literature, have set new benchmarks in the field. Moreover, Mechanical Design Agent (MDA) (Lu et al., 2024) demonstrates the use of LLMs for generating CAD models directly from text commands, highlighting advancements in automated design processes. In aviation, the RoBERT model, fine-tuned for domain-specific tasks, has achieved an impressive 82.8% accuracy in knowledge tasks (Nielsen et al., 2024), demonstrating the potential of LLMs in highly specialized fields.

**LLM Agents** The existing research on LLM-based agents can be categorised into single-agent and multi-agent systems. The single-agent systems utilize a single LLM for various applications such as task planning (Ge et al., 2024; Deng et al., 2024), API tool using (Schick et al., 2024; Parisi et al., 2022; Tang et al., 2023), web browsing (Nakano et al., 2021; Deng et al., 2024), and reasoning (Yao et al., 2024; Hao et al., 2023; Xiang et al., 2024; Yu et al., 2024; Ouyang et al., 2023). On the other hand, multi-agent systems such as Generative Agents (Park et al., 2023) simulates human behaviors by creating a town of 25 agents to study social understanding. CAMEL (Li et al., 2023) employs role-play techniques to study the behaviors and capabilities of a agents society. Some works explore the competitive multi-agent systems that involves agents debate, negotiate and competition to improve its performance in negotiation skills, question-answering (Fu et al., 2023; Du et al., 2023; Chan et al., 2023; Liang et al., 2023). ChatDev (Qian et al., 2023) developed a chat-powered software development framework in which specialized agents driven by large language models (LLMs).

## B.2 MORE BACKGROUND ON CLASSIC CONTROL

First, we give a detailed review of various standard control design objectives mentioned in our main paper.

- **Stability:** A poorly designed system can cause a system to go unstable, i.e. signals can grow unbounded. The practical consequence is that the system or device can be destroyed leading to financial loss or even loss of life. To avoid this, the controller $C(s)$ should be designed so that the feedback system is stable.

- **Fast Reference Tracking:** The controller should be designed so that the system output tracks the desired reference command. This involves various performance metrics but mainly the system should respond quickly to changes in the reference command.

- **Disturbance Rejection:** Disturbances $d(t)$ are external signals that affect the plant dynamics. For example, the a car with a cruise control system is affected by forces due to wind gusts and hills. The controller should be designed so that disturbances have small effect on tracking, i.e. result in small errors.

- **Actuator Limits:** The input signal generated by the controller should remain within allowable levels. For example the throttle (accelerator) on a car can only move by a certain amount. The command from the controller must remain within these allowable bounds.

- **Noise Rejection:** The feedback controller relies on a measurement. It is typically required that any measurement inaccuracies, e.g. noise, have small effect on tracking. Moreover, the noise should have a small effect on the control effort.

- **Robustness to Model Uncertainty:** As noted above, there exist some gaps between the model used for control design and the true systems that the controller is deployed on. The controller must be robust, i.e. insensitive, to model errors introduced by such gaps. Model uncertainty typically includes errors due to parameter variations and unmodeled dynamics.

Next, we review a few control-theoretic concepts that are crucial for classic control design. A fundamental requirement in most control engineering applications is **closed-loop stability** (Goodwin et al., 2001; Boyd & Barratt, 1991). For an LTI system with a transfer function $G(s)$, it is considered to be stable if all poles of the transfer function (i.e., the roots of the denominator) have negative real parts. The closed-loop stability means that the closed-loop transfer function from the reference signal $r(t)$ to the output signal $y(t)$ has to be stable. In the control language, the sensitivity function and the complementary sensitivity function are both required to be stable. Mathematically, we require all the roots of $1 + G(s)C(s) = 0$ to have strictly negative real parts.

### B.3 PERFORMANCE METRIC: SETTLING TIME

For a stable LTI system, the **settling time** $T_s$ is the time for the output to converge within $\pm 2\%$ of the steady-state value given that the input is a step function. Slightly different definitions are sometimes used, e.g. 5% or 1% settling times. Since PIDtune uses the 2% settling time as default, we also adopt 2% settling time in our study. The settling time is one main measure for the system speed of response. A shorter settling time typically indicates a faster response, which is desirable in many applications where rapid stabilization is critical (e.g., robotics, automotive systems, or process control). However, excessively fast responses can lead to undesirable side effects, such as overshoot or instability. Therefore, in ControlEval evaluation benchmark, we introduce three different response modes (fast, moderate, and slow) for first-order stable systems and second-order stable systems by requiring the settling time of the designed system within the reasonable predefined range. Currently, the settling time has been a standard evaluation metric for real control system design, and it can be effectively calculated through step_info command in Python control package.

### B.4 PERFORMANCE METRIC: PHASE MARGIN

For a stable LTI system, the **phase margin** is the amount of allowable variation in the phase of the plant before the closed-loop becomes unstable, and the **gain margin** is the amount of allowable variation in the gain of the plant before the closed-loop becomes unstable. As shown in Figure 4, phase margin can be determined from Bode plots. Phase margin is typically viewed as the most important robustness metric for classic control design. Since in real-world systems, phase shifts may occur due to delays, modeling inaccuracies, or external disturbances. Phase margin quantifies how much phase lag the system can withstand before instability arises. If the phase margin is too small (close to $0°$), the system is on the verge of instability and may exhibit oscillatory behavior. A larger phase margin (e.g., $30°$ to $60°$) typically indicates a more robust and stable system. Phase margin can be effectively calculated through margin command in Python control package.

### B.5 EXACT GUARANTEES IN CONTROLAGENT

To ensure the design of ControlAgent is indeed guaranteed to satisfy the requirements, we have employed a computation agent to perform exact and accurate evaluations (i.e., settling time and phase margin) and provide the feedback based on the evaluations for next iteration if the design is not successful. Therefore, once a design is successful, it has to pass the verification of the computation agent and the success is guaranteed.

## C    CONTROLAGENT WITH REAL-LIFE APPLICATIONS

To address the complexity of higher-order system control, we manually designed 50 stable and unstable higher-order systems along with their associated control requirements for ControlEval. The design process follows two structured methodologies:

1. **Mimicking Real-Life Applications**: Starting with the transfer functions of real-life systems, we designed higher-order systems by beginning with a dominant subsystem (e.g., first-order or second-order dynamics). We then introduced additional poles faster than the dominant poles to simulate higher-order dynamics such as unmodeled or negligible dynamics.

2. **Diverse Pole Configurations**: In this setting, we included systems where no single pole is dominant, achieved by randomly sampling pole positions such that all poles significantly contribute to the system's behavior. In this case, human experts ensured that the resulting systems remained controllable and adhered to practical design requirements, such as there existing a controller for the designed system to achieve desired settling time and phase margin.

This dual design approach ensures that the higher-order systems in ControlEval are representative of a wide range of real-world applications. To further enhance ControlEval's representativeness, we have added a new category of 10 higher-order tasks derived from real-life application contexts along with their evaluation results in Section C.1. Additionally, we apply the ControlAgent to implement a controller for an actual hardware system (DC motor) C.2, showcasing how the ControlAgent successfully meets hardware requirements and facilitates effective controller design.

### C.1    CONTROLAGENT WITH REAL-LIFE APPLICATIONS

To demonstrate the practical utility of ControlAgent in real-world applications, we present ten representative dynamical systems. The models for these systems are sourced from existing literature, with detailed descriptions available in the respective references. The design specifications for these systems are generated randomly, guided by established heuristics commonly used in control system design. These heuristics include $50^o \leq PM \leq 90^o$ and nominal settling time obtained from the system parameters such as damping ratio and natural frequency. We have tested the ControlAgent for the following practical systems [3]:

1. **Laser Printer Positioning system (Dorf & Bishop, 2007):** A laser printer prints using the precise position control of a laser onto the printing surface based on the input from the computer. The system dynamics for the laser printer position control is specified as follows:
$$T(s) = \frac{4(s+50)}{s^2 + 30s + 200}$$
Given the nature of the application, the reasonable design specs needs to be very demanding. In our study, we use the settling time, $T_s \leq 0.36$ and minimum phase margin of $74.24^o$.

2. **Space Station Orientation Control system (Dorf & Bishop, 2007):** The space station orientation control system can be modeled as a second order system given as follows:
$$T(s) = \frac{20}{s^2 + 20s + 100}$$
The design requirements were set at $76.22^o$ as minimum phase margin and $T_s \leq 0.64$.

3. **Vehicle Steering Control system (Dorf & Bishop, 2007):** One of the most commonly utilized system in real-world is the vehicle steering control problem. The vehicle steering control system can be represented as follows:
$$T(s) = \frac{1}{s(s+12)}$$

---

[3]The dynamical systems for different real-life applications are scaled in different time units, the exact settling time unit for each dynamical system can be found in the corresponding references.

The safety critical nature of this system requires a robust design specifications and fast response resulting in the minimum phase margin requirement of $56.98^o$ and settling time $T_s \leq 0.58$.

4. **Antenna Azimuth Control system (Nise, 2020):** Often times large antennas are deployed to receive satellite signals and they must accurately track the satellite as it moves across the sky. One such system can be modeled as follows:

$$T(s) = \frac{20.83}{s^2 + 101.7s + 171}$$

with the required design specification given by a minimum phase margin of $82.95^o$ to ensure stability to disturbances such as wind gusts along side a reasonable settling time requirement of $T_s \leq 1.57$.

5. **Autonomous Submersible Control system (Kuo, 1987):** The depth control system for an autonomous underwater vehicle is modeled as follows:

$$T(s) = \frac{-0.13(s + 0.44)}{s^2 + 0.23s + 0.02}$$

The settling time requirements for such systems falls under the category of slow systems reported in this work. The controller design specs for this system were $T_s \leq 41.49$ and the minimum phase margin of $69.49°$.

6. **Aircraft Pitch Control System (University of Michigan & Simulink):** The pitch dynamics of a commercial Boeing aircraft are given by the following transfer function:

$$T(s) = \frac{1.151s + 0.1774}{s^3 + 0.739s^2 + 0.921s}$$

The commercial aircraft is stable by design and thus typically falls in the category of large settling times. The design requirements are thus selected to be $T_s \leq 33.58$ and the minimum phase margin of $53.92°$.

7. **Missile yaw control system (Dorf & Bishop, 2007):** The yaw acceleration control system for a bank-to-turn missile is given by the following transfer function:

$$T(s) = \frac{-0.5(s^2 + 2500)}{(s - 3)(s^2 + 50s + 1000)}$$

The design requirements are thus selected to be $T_s \leq 3.95$ and the minimum phase margin of $63.43°$.

8. **Helicopter Pitch Control system (Dorf & Bishop, 2007):** The dynamics of a helicopter control system that utilizes an automatic control loop alongside a pilot stick control is given as follows:

$$T(s) = \frac{25(s + 0.03)}{(s + 0.4)(s^2 - 0.36s + 0.16)}$$

The design specifications use for the pitch control are $T_s \leq 30.36$ and the minimum phase margin of $66.81°$.

9. **Speed Control of a Hard Disk Drive:** In a hard disk drive, data is stored in tracks on spinning magnetic disks. A voice coil motor (VCM) moves the read/write head to the desired track and maintains its position during read/write operations. The dynamic behavior of the system, from the voltage input $u$ to the position $y$ of the read/write head relative to the track center, can be approximated by the following fourth-order transfer function:

$$T(s) = \frac{-0.1808s^4 - 0.5585s^3 + 0.4249s^2 - 8.625s + 135.1}{s^4 + 0.2046s^3 + 8.932s^2 + 0.1148s + 0.007285}$$

The design specifications use for the hard disk drive control are $T_s \leq 88.11$ and phase margin larger than $52.22°$.

| Methods | Success Rate (SR, %) |
|---------|:-------------------:|
| **Zero-shot** | 10 |
| **Zero-shot CoT** | 0 |
| **Few-shot** | 20 |
| **Few-shot CoT** | 0 |
| **PIDtune** | 50 |
| **ControlAgent** | **100** |

Table 5: Success Rate (SR, %) of baseline methods and ControlAgent on the real-world systems. The best result is highlighted in bold. The results show that ControlAgent outperforms all other LLM-based and toolbox-based baselines, demonstrating its effectiveness and robustness in handling diverse real-world systems.

10. **High speed train control system (Dorf & Bishop, 2007):** Consider a high speed train similar to the French Train á Grande Vitesse (TGV) which speeds up to 186 miles per hour. In order to achieve such high speeds on tight curves, it uses a tilt control mechanism. The transfer function of the tilt mechanism is given as follows:

$$T(s) = \frac{12}{s(s+10)(s+70)}$$

with the desired specifications for its control given by the settling time $T_s \leq 2.08$ and a minimum phase margin of $74.40^o$.

### C.2 CONTROLAGENT WITH HARDWARE IN THE LOOP

The usage of the ControlAgent is further demonstrated by its practical application for the position control of a DC motor. We consider the following nominal model of the DC motor:

$$T(s) = \frac{K}{s\left((L_a s + R_a)(Js + b) + K_\tau K_v\right)}$$

where the nominal values for these parameters can be found in Table 6.

| Parameter | $K$ | $K_\tau$ | $K_v$ | $L_a$ | $R_a$ | $J$ | $b$ |
|-----------|-----|----------|-------|-------|-------|-----|-----|
| **Value** | 0.2036 | 0.0533 | 0.0533 | 0.000975 | 4.465 | 0.0001227 | 0.00005 |

Table 6: Parameter values for DC motor model.

We provide a nominal third order model of the motor to ControlAgent along with the design specs of $T_s \leq 1.2$ [4] and PM of $55^o$ to design a feasible loop shaping PID controller for the motor. Utilizing the iterative procedure, ControlAgent outputs the loop gains for a successful design. The proposed controller in then passed to the interface module which utilizes Matlab and Simulink frameworks to deploy the controller to the physical motor. The interface module simultaneously deploys the controller and also collects the feedback in real time thus achieving a hardware in loop architecture, an industry standard procedure for designing controllers for real world systems. The complete framework and the DC motor setup is represented in Fig. 7.

A performance comparison of the controllers generated by ControlAgent and the PIDTune baseline as well as the LLM based baselines was carried out for the DC motor setup. Given the motor model and the design specification of settling time and phase margin, the only controller competitive to the ControlAgent was generated by the PIDTune framework. The ControlAgent based controller results in the overshoot of 7.49% and the settling time of $T_s = 1.1253$ against the overshoot of 7.342% and $T_s = 3.0093$ for the PIDTune based controller. Clearly, the PIDTune generated controller fails to meet the design requirement for the settling time $T_s$.

A comparison of the trajectory tracking performance is showcased in Fig. 8. Here it is important to note that ControlAgent generated the desired controller without any additional information whereas

---

[4]For DC motor, the unit for the settling time is in second.

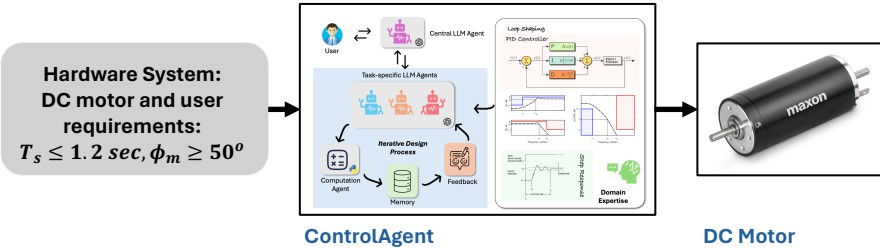

(a): ControlAgent with DC motor in real-life application

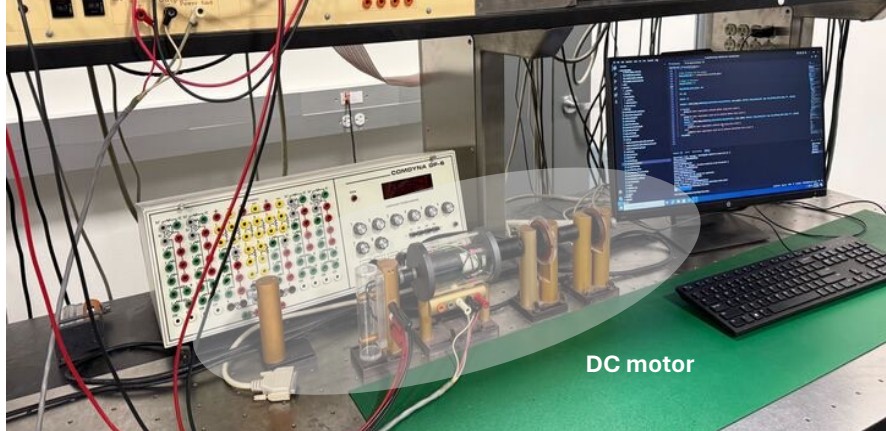

(b): DC motor at Lab

Figure 7: Implementation of ControlAgent on DC Motor Setup.

in case of PIDTune, in order to achieve a reasonable performance, we additionally provided the crossover frequency of the system. Here, we supplied the same crossover frequency to PIDTune as used by the ControlAgent loopshaping procedure. Additionally, observe that the real-world systems such as the DC motor does suffer from sensor noise issues as evident from the zoom-in of the motor trajectories in Fig. 8. A comparison of the control inputs generated by ControlAgent and PIDTune is provided in Fig. 10 which suggests that the control input computed by the two approaches remains with-in the saturation bounds and are comparable for the two controllers. A comparison of the feasible controllers generated by LLM based baselines is provided in Fig. 9. Clearly, none of the baselines managed to produce a controller that meets the design specifications. All the baselines not only fail the settling time requirements but also results in significant steady-state errors. This comparative study thus effectively highlight the usefulness of ControlAgent for real-world applications.

## D  MORE ON EXPERIMENTAL STUDY

### D.1  MORE ON THE EXPERIMENTAL SETUP

**LLM-based Baselines.**  We evaluate four LLM-based baseline approaches: zero-shot prompting, zero-shot Chain-of-Thought (CoT), few-shot, and few-shot CoT. For the few-shot baselines, we provide two demonstration examples tailored to the specific task type. For instance, in the case of first-order unstable system design, the few-shot setting includes two examples demonstrating successful controller designs for unstable first-order systems, along with the associated control design criteria. In the few-shot CoT setting, we further include detailed reasoning steps to illustrate the process of designing a successful controller for the given demonstration examples. A complete example of the few-shot CoT prompt is provided in Appendix E.2. All LLM-based baselines are implemented using GPT-4o, with model hyperparameter settings detailed in Table 7.

**Control Toolbox-based Baseline.**  We use the widely employed control design toolbox PIDtune as a baseline, which provides various settings for tuning PID controllers for linear and higher-order

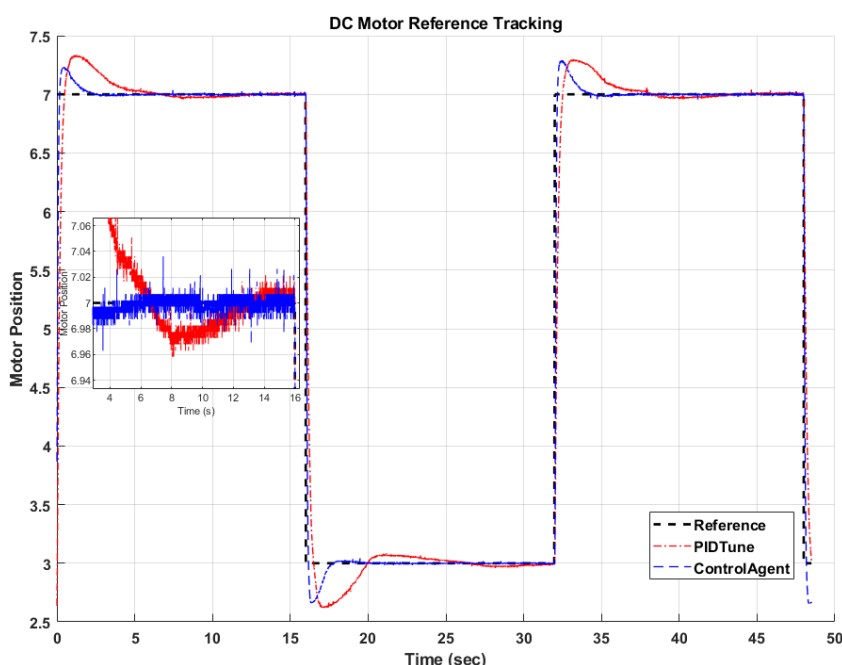

Figure 8: Comparison of ControlAgent with the baselines for the position control of DC Motor: ControlAgent vs the PIDTune baseline.

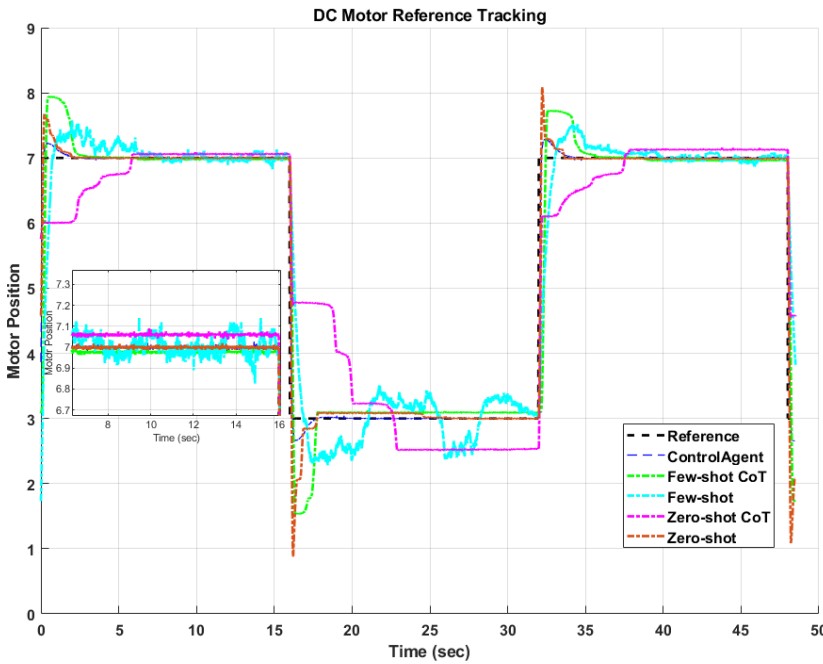

Figure 9: Comparison of ControlAgent with the baselines for the position control of DC Motor: ControlAgent vs the LLM based baselines.

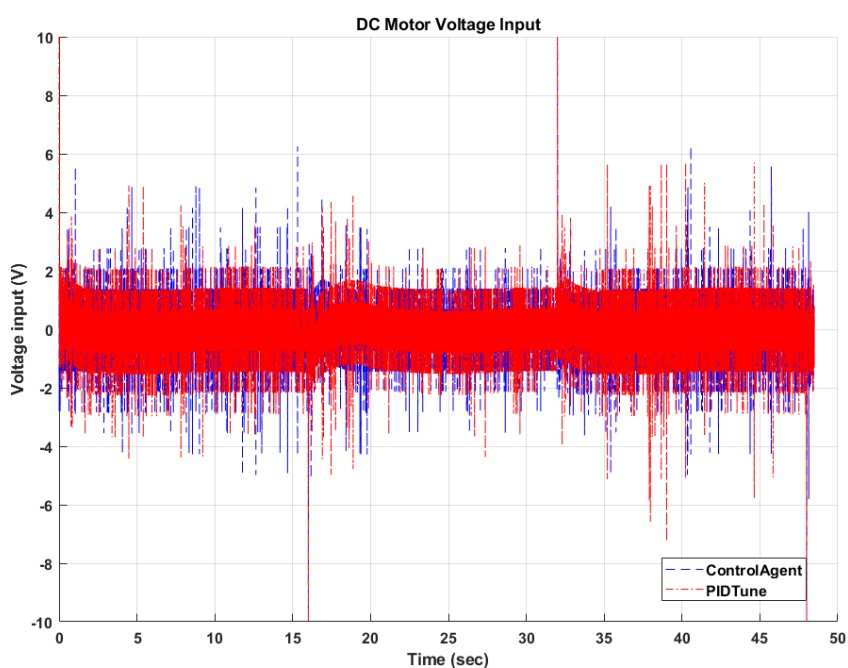

Figure 10: Voltage inputs for ControlAgent and PIDTune

systems. To ensure a fair comparison, we incorporate control domain knowledge to specify appropriate inputs, such as phase margin and crossover frequency, to PIDtune, enabling it to achieve the desired control design criteria in a single step. Specifically, we use two distinct configurations to establish PIDtune as a baseline:

For first and second-order systems (both stable and unstable), PIDtune is configured with the desired phase margin and open-loop crossover frequency. The phase margin is directly derived from the task requirements, while the crossover frequency is determined as $C/T_s$, where $C \in [3, 5]$ is a constant, and $T_s$ is the required settling time. For each trial, $C$ and $T_s$ are randomly sampled within their specified ranges to align with the design specifications.

For higher-order systems, the relationship between the crossover frequency and the settling time $T_s$ is not clear in general. As a result, only the phase margin requirement is supplied to PIDtune to optimize the PID gains.

**Parameter Setup for ControlAgent.**  In ControlAgent, the maximum number of iterations is configured based on the difficulty of the task. For relatively straightforward tasks, such as stable first-order and stable second-order systems, the maximum number of iterations is set to 10. For more challenging tasks, including first-order systems with delay, unstable first-order systems, and unstable second-order systems, the maximum iteration count is increased to 20. Finally, for the most difficult tasks, such as higher-order systems, the maximum number of iterations is set to 30 to allow for additional refinement and ensure that the desired control criteria are met.

**LLM hyperparameters.**  Table 7 presents the hyperparameter settings used for each LLM model in this study, including model versions, temperature settings, and maximum token limits.

D.2   EVALUATION METRIC

**Average Successful Rate**  Suppose for each control task, our evaluation dataset consists of $N$ sample systems and the associated predefined criteria, such as stability, phase margin, settling time. Let $S_{i,j}$ denote the outcome of the $j$-th trial for the $i$-th system, where $S_{i,j} = 1$ if the design is

| Model | Hyperparameters |
|-------|-----------------|
| GPT-4o | model = `gpt-4o-0806`, temperature = 0, max_tokens = 1024 |
| GPT-4-turbo | model = `gpt-4-turbo`, temperature = 0, max_tokens = 1024 |
| GPT-3.5-turbo | model = `gpt-3.5-turbo-0125`, temperature = 0, max_tokens = 1024 |
| Claude-3.5 | model = `claude-3-5-sonnet-20240620`, temperature = 1, max_tokens = 1024 |
| Gemini Pro 1.5 | model = `gemini-1.5-pro`, temperature = 1, max_tokens = 8192 |

Table 7: Hyperparameter configurations for each LLM model used in this study.

successful and $S_{i,j} = 0$ otherwise. The averaged successful rate for trial $j$ is computed as

$$\text{ASR}_j = \frac{1}{N} \sum_{i=1}^{N} S_{i,j},$$

and the overall ASR across all $T$ trials is given by

$$\text{ASR} = \frac{1}{T} \sum_{j=1}^{T} S_j.$$

This metric provides insight into the average performance of the controller design over multiple trials for each system, reflecting its consistency.

**Aggregate Successful Rate** This metric evaluates success on a system-by-system basis, where a system is considered successfully designed if at least one of the $T$ independent trials results in a successful design. Specifically, the aggregated success for system $i$ is:

$$\text{AgSR}_i = \begin{cases} 1 & \text{if } \sum_{j=1}^{T} S_{i,j} > 0, \\ 0 & \text{otherwise} . \end{cases} \tag{3}$$

The overall AgSR across all systems is then computed as

$$\text{AgSR} = \frac{1}{N} \sum_{i=1}^{N} \text{AgSR}_i.$$

This metric is generally higher than the ASR since it only requires one successful trial per system for the entire system to be considered a success. It reflects the best controller design for each system, providing a more lenient evaluation of the controllers overall performance.

In the experimental study, for each control task, we have $N = 50$ and we ran each control tasks for five trails ($T = 5$) for both ControlAgent and baseline methods.

**Metric** $pass@k$ To provided a more robust evaluations metric, we employed $pass@k$ metric introduced in (Chen et al., 2021). Specifically, we ran ControlAgent $n \geq k$ trials per task, count the successful designs $c \leq n$ which satisfy the pre-defined requirements, and calcualte the following unbiased estimator:

$$pass@k := \mathbb{E}_{\text{trials}} \left[ 1 - \frac{\binom{n-c}{k}}{\binom{n}{k}} \right]. \tag{4}$$

In this work, we choose $n = 5$ and $k = \{1, 3, 5\}$. Noticing that when $k = n$, this metric is the same the AgSR metric defined above.

### D.3 MORE DETAILS ON THE MEMORY MODULE

In ControlAgent, we address these limitations by selectively storing and providing only the essential historical information to the LLMs. Specifically, we retain key data such as design parameters,

---

**Algorithm 1:** Iterative Controller Design Process of ControlAgent

---

**Input:** User requirements $\mathcal{R}$, Maximum iterations $N_{\max}$
**Output:** Designed controller $C$
Initialize memory buffer: $\mathcal{M} \leftarrow \emptyset$;
Initialize feedback: $\mathcal{F}_0 = \{\}$;
**Task Assignment:** $\mathcal{A}_c$ assigns the task to $\mathcal{A}_{\text{spec}}$ based on $\mathcal{R}$: $\mathcal{A}_{\text{spec}} \leftarrow \text{AssignTask}(\mathcal{A}_c, \mathcal{R})$
**for** $k = 1$ **to** $N_{max}$ **do**
    Generate input prompt: $\mathcal{P}_k \leftarrow \text{GenPrompt}(\mathcal{E}_{\text{spec}}, \mathcal{R}, \mathcal{M}, \mathcal{F}_{k-1})$;
    LLM agent generates controller: $C_k \leftarrow \mathcal{A}_{\text{spec}}(\mathcal{P}_k)$;
    Update memory buffer: $\mathcal{M} \leftarrow \mathcal{M} \cup \{C_k\}$;
    Python agent $\mathcal{A}_p$ evaluates $C_k$ and computes performance $P_k$;
    **if** $P_k$ *satisfies* $R$ **then**
        **return** Successfully designed controller $C_k$;
    **else**
        Generate feedback: $\mathcal{F}_k \leftarrow \text{GenFeedback}(P_k, R)$;

**return** No successful controller is found;

---

performance metrics, and feedback from previous iterations, while excluding unnecessary details from the LLMs responses. This ensures that the memory buffer remains compact and efficient. For example, here is an illustration of two historical designs stored in the memory buffer for one specific task:

---

**ControlAgent Memory Module Demonstration**

### Design 1
Parameters: omega_L=5.5, beta_b=3.162, beta_l=3.162
Performance: phase_margin=15.89, settling_time = 3.20, steadystate_error=0.0
Feedback: Phase margin should be at least 52.82 degrees.

### Design 2
Parameters: omega_L=6.5, beta_b=4.0, beta_l=4.0
Performance: phase_margin=30.06, settling_time=3.60, steadystate_error=0.0
Feedback: Phase margin should be at least 52.82 degrees.

---

Only the above summarized history is fed into the LLM for the next iteration. This strategy allows the LLM to focus on refining the design based on the key feedback and performance metrics, without exceeding the context window limitations. From our observations, this approach effectively prevents context window overflow while maintaining the iterative design process. Additionally, it ensures memory efficiency by retaining only the critical information required to improve upon previous designs.

## D.4 MORE EXPERIMENTAL RESULTS

### D.4.1 CONTROLAGENT WITH OPEN-SOURCE MODEL

In this section, we evaluate the performance of ControlAgent on a more accessible open source LLM backbone, Llama-3.1-70b (Dubey et al., 2024). In particular, we implemented ControlAgent with Llama-3.1-70b on four tasks: first-order stable systems with fast, moderate, and slow response modes, and another harder control problem with higher-order system design. We report $pass@k$ with $k = \{1, 3, 5\}$ and the average number of iterations per task in Table 8.

From Table 8, it can be seen that ControlAgent with Llama-3.1-70b is also effective for simpler, first-order control tasks but faces challenges with more complex, higher-order systems. The $pass@1$ rate is only 0.300, indicating that the model struggles to solve the problem on the first attempt. The $pass@3$ and $pass@5$ rates improve to 0.446 and 0.480, respectively, but still remain below 50%, suggesting that the task is considerably more challenging for the model.

| System Type | 1st-ord stb fast | 1st-ord stb moderate | 1st-ord stb slow | Hgr-ord |
|---|---|---|---|---|
| $pass@1$ | 0.927 | 1.000 | 0.996 | 0.300 |
| $pass@3$ | 1.000 | 1.000 | 1.000 | 0.446 |
| $pass@5$ | 1.000 | 1.000 | 1.000 | 0.480 |
| **iteration #** | 3.053 | 2.016 | 2.824 | 24.5 |

Table 8: Performance of ControlAgent with Llama-3.1-70b.

In addition, the average number of iterations required for first-order stable systems is relatively low, with moderate response mode requiring the fewest iterations (2.016), followed by slow (2.824) and fast (3.053). In contrast, the higher-order system requires a significantly higher average number of iterations (24.5), reflecting the increased complexity and difficulty of the task.

The results suggest that while the ControlAgent with Llama-3.1-70b performs well on simpler, first-order stable systems, it struggles with more complex, higher-order control problems. This indicates a performance gap between Llama-3.1-70b and current state-of-the-art models such as GPT-4o, which may be more adept at handling such complex tasks.

### D.4.2    EVALUATION RESULTS WITH METRIC $pass@k$

In this section, we report the metric $pass@k$ with $k = \{1, 3, 5\}$ of ControlAgent with GPT-4o as the LLM backbone. The results are shown in Table 9.

| System Type | 1st-ord stb | | | 2nd-ord stb | | | 1st-ord ustb | 2nd-ord ustb | w/ dly | Hgr-ord |
|---|---|---|---|---|---|---|---|---|---|---|
| **Response Mode** | fast | moderate | slow | fast | moderate | slow | (-) | (-) | (-) | (-) |
| $pass@1$ | 1.00 | 1.00 | 1.00 | 1.00 | 1.00 | 1.00 | 0.98 | 0.91 | 0.97 | 0.82 |
| $pass@3$ | 1.00 | 1.00 | 1.00 | 1.00 | 1.00 | 1.00 | 1.00 | 0.99 | 1.00 | 0.95 |
| $pass@5$ | 1.00 | 1.00 | 1.00 | 1.00 | 1.00 | 1.00 | 1.00 | 1.00 | 1.00 | 0.96 |

Table 9: Metric $pass@k$ of ControlAgent with GPT-4o as LLM backbone.

It can be seen that ControlAgent is able to achieve high $pass@k$ rates, even for complex tasks, highlights its robustness and effectiveness in handling diverse control problems.

### D.4.3    MORE RESULTS ON THE ITERATION NUMBER

In Table 10, we present the average number of iteration number (sample size) for each type of control problem:

| System Type | 1st-ord stb | | | 2nd-ord stb | | | 1st-ord ustb | 2nd-ord ustb | w/ dly | Hgr-ord |
|---|---|---|---|---|---|---|---|---|---|---|
| **Response Mode** | fast | moderate | slow | fast | moderate | slow | (-) | (-) | (-) | (-) |
| iteration # | 2.74 | 1.78 | 2.19 | 2.37 | 2.64 | 3.72 | 3.90 | 5.72 | 9.91 | 9.56 |

Table 10: Metric $pass@k$ of ControlAgent with GPT-4o as LLM backbone.

The results in Table 10 provide an estimate of the sample efficiency of ControlAgent. It is evident that ControlAgent demonstrates considerable efficiency across all control problems, with fewer than 10 iterations required on average.

### D.4.4    MORE RESULTS ON THE ROBUSTNESS OF CONTROLAGENT

ControlAgent addresses uncertainty through the core design principle via loop-shaping. To demonstrate this, we evaluated the disk margins of the designed controllers from ControlAgent. Disk margins provide a comprehensive robustness measure by simultaneously addressing both gain margin and phase margin uncertainties, encapsulating the effects of non-parametric unmodeled dynamics (Seiler et al., 2020). These margins are particularly suitable for ensuring robust performance under a wide range of operating conditions.

Table 11 demonstrates the averaged disk margins (including gain margins and phase margins) for controllers designed by ControlAgent, showing robust stability across various control problems.

| System Type | Disk Margin | Phase Margin | Gain Margin |
|---|---|---|---|
| **1st-ord stb fast** | $1.6184 \pm 0.0238$ | [-77.7608, 77.7608] | [0.1074, Inf] |
| **1st-ord stb moderate** | $1.9833 \pm 0.0046$ | [-89.4855, 89.4855] | [0.0045, Inf] |
| **1st-ord stb slow** | $2 \pm 0$ | [-90, 90] | [0, Inf] |
| **2nd-ord stb fast** | $1.2553 \pm 0.0255$ | [-63.9961, 63.9961] | [0.2317, 4.6367] |
| **2nd-ord stb moderate** | $1.2852 \pm 0.0304$ | [-65.1729, 65.1729] | [0.2211, 4.9703] |
| **2nd-ord stb slow** | $1.4498 \pm 0.0134$ | [-71.7560, 71.7560] | [0.1609, 6.5571] |
| **1st-ord ustb** | $1.0695 \pm 0.0976$ | [-55.3823, 55.3823] | [0.3157 4.4090] |
| **2nd-ord ustb** | $0.9646 \pm 0.0257$ | [-51.2628, 51.2628] | [0.3532, 2.9645] |
| **w/ dly** | $0.6628 \pm 0.2021$ | [-35.2660, 35.2660] | [0.5492, 2.3591] |
| **Hgr-ord** | $1.1058 \pm 0.0693$ | [-56.8894, 56.8894] | [0.3014, Inf] |

Table 11: Disk Margin ControlAgent.

These results show that ControlAgent maintains adequate robustness margins even under varying conditions, reinforcing its capability to handle non-parametric uncertainties effectively.

### D.4.5 FAILURE MODES ANALYSIS

In this section, we identified several failure modes in ControlAgent for higher-order system design, each revealing challenges in reasoning and parameter adjustment strategies:

1. **Calculation Errors**: One notable failure occurred with a marginally unstable system featuring a double integrator. The LLM incorrectly calculated the minimum loop bandwidth as "*The fastest unstable pole is at 0, so we initially chose $\omega_L = 2.5 \times 0 = 2.5$.*" This calculation was incorrect and did not align with proper design principles.

2. **Incomplete Parameter Adjustments**: The LLM often adjusted only two parameters ($\omega_L$ and $\beta_l$), neglecting $\beta_b$, which is crucial for balancing the settling time and phase margin. For example, in one design, the final parameters were $\omega_L = 60$, $\beta_b = 0.8$, and $\beta_l = 1000$, with $\beta_b$ remaining unchanged throughout iterations. This limited adjustment scope hindered optimal design.

3. **Hallucination Errors**: Another failure involved misidentifying the dominant pole. In one instance, the LLM incorrectly identified -50 as the dominant pole instead of the actual dominant poles at $-2.1 \pm 2.142$: "*The poles at -50 and $-2.1 \pm 2.14242853j$ suggest a relatively fast response due to the dominant pole at -50.*" This misunderstanding led to incorrect design decisions.

These examples highlight key areas where the LLM's reasoning and parameter optimization strategies need improvement. Addressing these failure modes is a priority for future iterations of the framework.

### D.4.6 MORE RESULTS ON AGSR

Table 12 shows the AgSR of ControlAgent and various baseline methods on the ControlEval benchmark. The best results for each task are highlighted in bold. Our key observations are given below.

**ControlAgent consistently outperforms all baseline methods.** ControlAgent achieves significantly higher AgSR across all control tasks compared to both LLM-based and traditional toolbox-based baselines. This superior performance is evident not only for simpler first-order and second-order stable systems but also for more complex cases, such as unstable and higher-order systems. While PIDtune performs well for first-order and second-order stable systems, as well as first-order systems with time delay, it struggles in more challenging scenarios like first-order unstable, second-order unstable, and higher-order systems. In these cases, ControlAgent's effectiveness becomes more apparent, especially as the complexity of the problem increases.

Its important to note that because of the inherent randomness in the answers generated by LLMs, individual runs of ControlAgent may yield different outcomes. However, when looking at aggregate results across multiple iterations, ControlAgent achieves 100% success across all design problems except for higher-order systems, where it still achieves an impressive 96%. While this falls short of 100%, it is a significantly better result than any other toolbox-based method and LLM-based baselines, and given the difficulty of higher-order system design, this accuracy is very promising.

| System Type | 1st-ord stb | | | 2nd-ord stb | | | 1st-ord ustb | 2nd-ord ustb | w/ dly | Hgr-ord |
|---|---|---|---|---|---|---|---|---|---|---|
| Response Mode | fast | moderate | slow | fast | moderate | slow | (-) | (-) | (-) | (-) |
| **Zero-shot** | 36.0 | 56.0 | 32.0 | 36.0 | 46.0 | 26.0 | 14.0 | 2.0 | 48.0 | 6.0 |
| **Zero-shot CoT** | 66.0 | 12.0 | 2.0 | 34.0 | 48.0 | 40.0 | 14.0 | 2.0 | 38.0 | 24.0 |
| **Few-shot** | 32.0 | 58.0 | 50.0 | 44.0 | 38.0 | 38.0 | 28.0 | 52.0 | 36.0 | 28.0 |
| **Few-shot CoT** | 36.0 | 66.0 | 68.0 | 26.0 | 40.0 | 60.0 | 18.0 | 62.0 | 60.0 | 36.0 |
| **PIDtune** | 94.0 | 100.0 | 100.0 | 98.0 | 100.0 | 100.0 | 68.0 | 42.0 | 100.0 | 50.0 |
| **ControlAgent** | 100.0 | 100.0 | 100.0 | 100.0 | 100.0 | 100.0 | 100.0 | 100.0 | 100.0 | 96.0 |

Table 12: AgSR (%) of baseline methods and ControlAgent.

### D.4.7 ASR vs AsGR

In Figure 11, we present the ASR and AgSR results for ControlAgent and other baseline methods across first-order and second-order stable systems. ControlAgent consistently outperforms all other methods in both ASR and AgSR metrics. While PIDTune delivers results comparable to ControlAgent in these cases, all other methods show significantly lower ASR values. However, their AgSR is noticeably higher than their ASR, which can be attributed to the inherent randomness in LLM-generated responses. This highlights the benefit of aggregate success, where the variability of LLM outputs improves performance over multiple iterations.

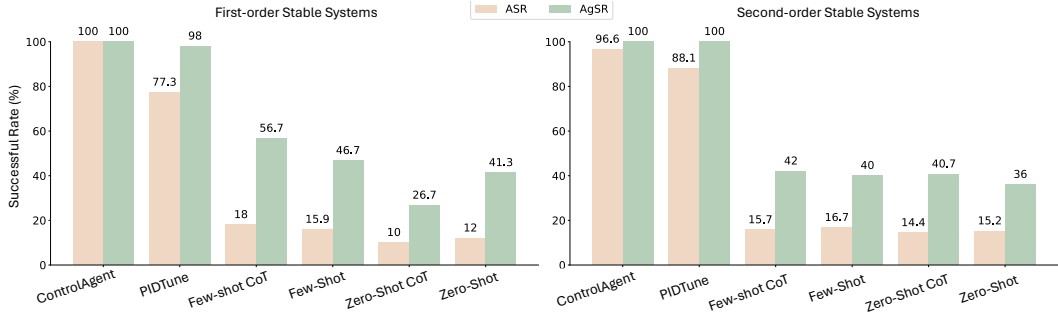

Figure 11: ASR and AgSR for different methods on first-order and second-order stable systems (averaged successful rate for three response speed types).

As we shift to more challenging unstable systems, Figure 12 further illustrates ControlAgent's increasing effectiveness. In these cases, the performance of the other methods declines sharply compared to their results on stable systems, as the complexity of the control design increases. Interestingly, this is where the limitations of randomness in LLMs become evident—both ASR and AgSR remain low for other methods, as even multiple iterations fail to improve their performance meaningfully. In contrast, few-shot-cot outperforms PIDTune in these harder scenarios, showcasing the potential of in-context learning when dealing with complex control tasks.

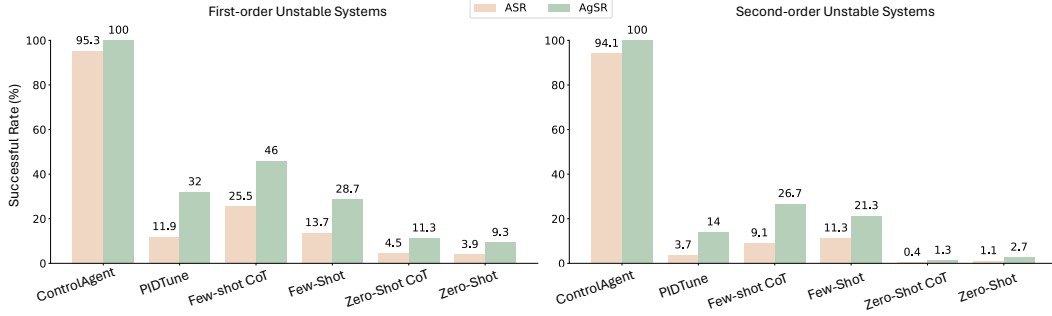

Figure 12: ASR and AgSR for different methods on first-order and second-order unstable systems (averaged successful rate for three response speed types).

In Figure 13, we examine first-order systems with delay and higher-order systems. For the first-order systems with delay, the performance trends mirror those of stable systems, with ControlAgent and PIDtune maintaining their strong lead. However, for higher-order systems, the accuracy of all methods, except ControlAgent, drops significantly. Despite the increased complexity of higher-order systems, ControlAgent continues to demonstrate impressive performance, highlighting its ability to handle even the most difficult control design problems. This underscores ControlAgent's robustness and adaptability, making it a clear leader among the tested methods.

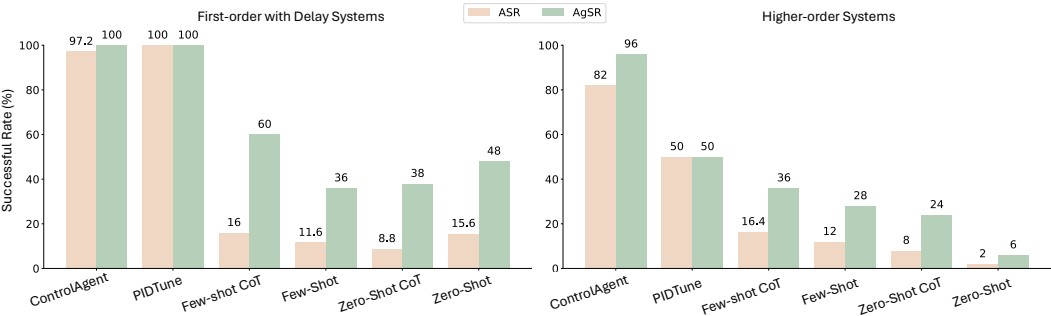

Figure 13: ASR and AgSR for different methods on first-order with delay and higher order systems.

## D.5 GAIN MARGIN CONSIDERATION

It is also important to highlight that control design can be evaluated using various metrics, with settling time and phase margin being two of the key ones we used. In this section, we further explore the effectiveness of ControlAgent by evaluating another important robustness metric: gain margin. For this analysis, we focus on comparing the designs produced by ControlAgent and PIDtune—our two best-performing methods—both of which already satisfy the settling time and phase margin requirements. We then examine how well their designs perform in terms of gain margin.

Typically, a good control design should have a gain margin within $\pm 6$ dB to ensure robustness against model uncertainty. As shown in Table 14, ControlAgent consistently outperforms PIDtune across nearly all scenarios, with the only exception being the first-order system with time delay. Notably, when comparing Table 14 with Table 2, we observe an interesting trend: every ControlAgent design that meets the settling time and phase margin requirements also inherently satisfies the gain margin criterion. This is a crucial result, as gain margin is typically another requirement that control engineers strive to achieve for robust designs.

In contrast, PIDtune's performance drops significantly when evaluated by gain margin, especially in more complex systems such as unstable and higher-order systems. This widening performance gap underscores ControlAgent's superior ability not only to meet the basic design requirements but also to inherently balance robustness, making its designs more resilient to model uncertainties.

| System Type | 1st-ord stb | | | 2nd-ord stb | | | 1st-ord ustb | 2nd-ord ustb | w/ dly | Hgr-ord |
|---|---|---|---|---|---|---|---|---|---|---|
| Response Mode | fast | moderate | slow | fast | moderate | slow | (-) | (-) | (-) | (-) |
| PIDtune | 56.0 | 90.4 | 86.4 | 65.2 | 54.8 | 75.2 | 0.0 | 0.0 | 100.0 | 16.0 |
| ControlAgent | 100.0 | 100.0 | 100.0 | 100.0 | 98.8 | 90.8 | 97.2 | 96.8 | 97.2 | 82.0 |

Table 14: ASR of PIDtune and ControlAgent on ControlEval for various system types with gain margin as an extra requirement.

## D.6 EVOLUTION OF CONTROLAGENT DESIGN

In this subsection, we analyze how ControlAgent's performance evolves over iterations to achieve the desired design. Figure 14 illustrates the simplest case, where ControlAgent is tasked with controlling first-order stable systems. Initially, all three response modes experience substantial settling time errors. However, these errors decrease rapidly, particularly in the slow scenario, which is the easiest to manage. This is because a slower system response reduces sensitivity to phase margin

violations, making it easier to meet performance requirements. The moderate scenario, however, requires a few additional iterations to reach the design objectives. As shown in the right plot of Figure 14, the fast scenario presents the most challenge, with a significant phase margin error early on. However, as the iterations progress, ControlAgent successfully reduces both the phase margin and settling time errors, demonstrating its ability to optimize system performance even in scenarios where there is a clear trade-off between performance and robustness.

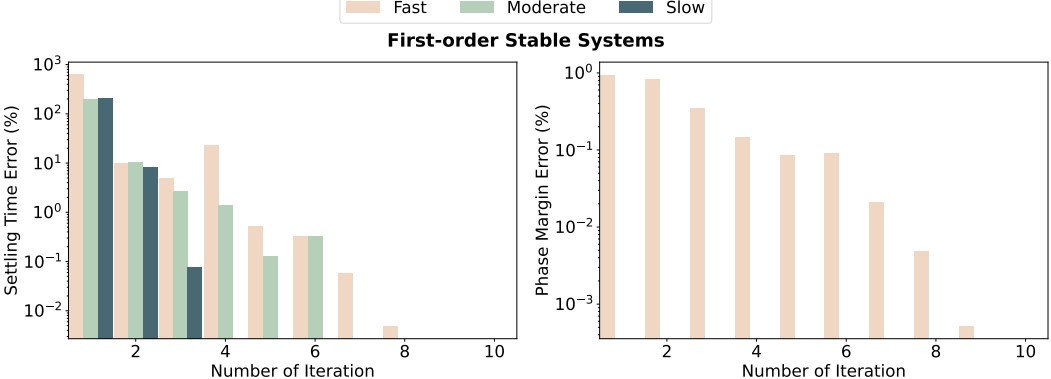

Figure 14: The behavior of ControlAgent across iterations for first-order stable systems. The left figure shows the change in settling time error, while the right figure tracks the phase margin error, both improving over iterations.

For each scenario, the design requires that the settling time $T_s$ falls within a specified range, $T_s \in [T_{s_{\min}}, T_{s_{\max}}]$, and that the phase margin $\phi$ meets or exceeds a minimum threshold, $\phi \geq \phi_{\min}$. During each iteration, if the settling time is within this range, the steady-state error is set to zero. Similarly, if the designed phase margin exceeds the required minimum, the phase margin error is set to zero. However, if the designed settling time $T_s$ exceeds $T_{s_{\max}}$, the steady-state error is computed as:

$$\text{Settling Time Error (\%)} = \frac{T_s - T_{s_{\max}}}{t_{s_{\max}} - T_{s_{\min}}} \times 100$$

Conversely, if the settling time is below $T_{s_{\min}}$, the error is calculated as:

$$\text{Settling Time Error (\%)} = \frac{T_{s_{\min}} - T_s}{T_{s_{\max}} - T_{s_{\min}}} \times 100$$

If the designed phase margin falls below $\phi_{\min}$, the phase margin error is determined as:

$$\text{Phase Margin Error (\%)} = \frac{\phi - \phi_{\min}}{\phi_{\min}} \times 100$$

Moving on to more complex systems, Figure 15 shows how ControlAgent's performance evolves when dealing with first-order unstable systems. In this case, both steady-state and phase margin errors start out large but gradually decrease as the agent iterates. Since unstable systems are more difficult to design for, ControlAgent is given up to 20 iterations to find a solution. This extended process highlights the agents ability to progressively refine system performance and robustness, improving both settling time and phase margin simultaneously, even in more challenging environments.

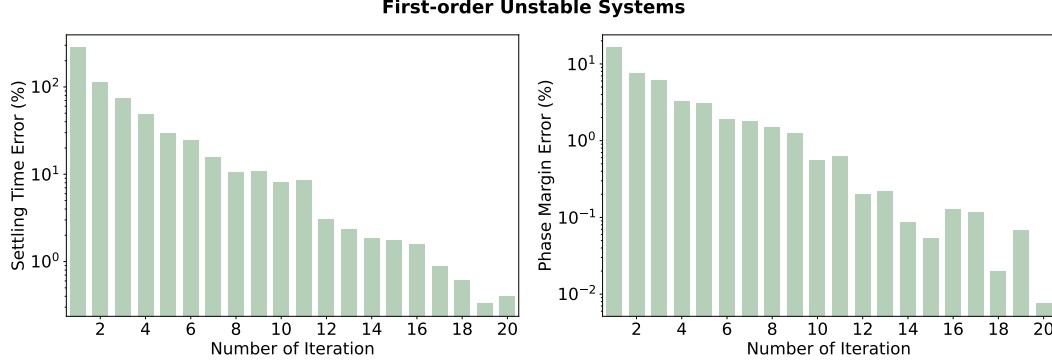

Figure 15: The behavior of ControlAgent across iterations for first-order unstable systems. The left figure shows the change in settling time error, while the right figure tracks the phase margin error, both improving over iterations.

The complexity increases further when ControlAgent tackles second-order stable systems, as shown in Figure 16. In the fast scenario, while the agent reduces settling time error within the first four iterations, this comes at the cost of a temporary increase in phase margin error, reflecting the trade-off between performance and robustness. However, with additional refinement, ControlAgent manages to bring the phase margin back within acceptable limits while still maintaining system performance. A similar trend is seen in the moderate scenario. However, unlike with first-order stable systems, the slow scenario poses the greatest challenge for second-order systems.This is because slower dynamics, while generally making a system more stable, reduce the systems responsiveness to control inputs. As a result, the system can become less robust over time, making it harder for the controller to maintain the required phase margin.

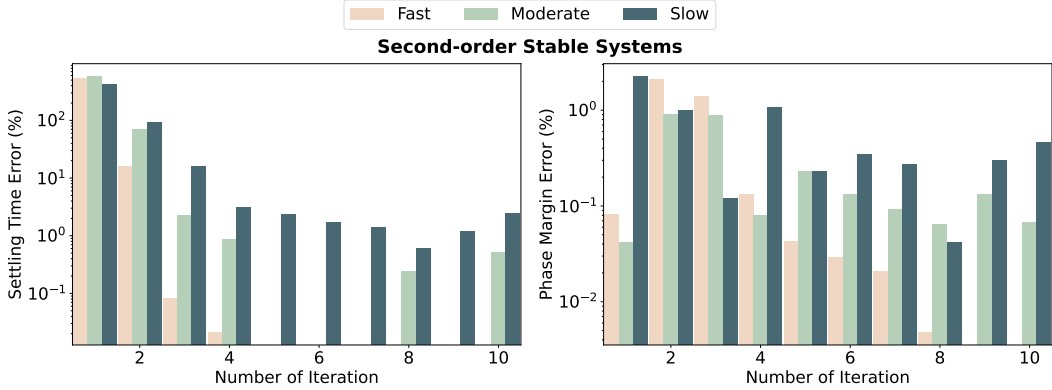

Figure 16: The behavior of ControlAgent across iterations for second-order stable systems. The left figure shows the change in settling time error, while the right figure tracks the phase margin error, both improving over iterations.

# E PROMPT DESIGN

## E.1 FULL PROMPTS FOR CONTROLAGENT

In this section, we provide the full prompt for ControlAgent, including the prompt for central agent to distribute the control tasks and a sample prompt for the task-specific agent for designing controllers of first-order stable system.

---

**Central Agent Prompt**

You are an expert control engineer tasked with analyzing the provided control task and assigning it to the most suitable task-specific agent, each specializing in designing controllers for specific system types.

First, analyze the dynamic system to identify its type, such as a first-order stable system, second-order unstable system, first-order with time delay, higher-order system, etc. Based on this analysis, assign the task to the corresponding task-specific agent that specializes in the identified system type.

Here are the available task-specific agents:

- **Agent 1**: First-order stable system

- **Agent 2**: First-order unstable system

- **Agent 3**: Second-order stable system

- **Agent 4**: Second-order unstable system

- **Agent 5**: First-order system with time delay

- **Agent 6**: Higher-order system

Ensure the selected agent can effectively tailor the control design process.

---

**Central Agent Response Instruction**

## Response Instructions:

Your response should strictly follow the JSON format below, containing three keys: 'Task Requirement' and 'Task Analysis', and 'Agent Number':
- **Task Requirement**: Summarize the task requirements, including the system dynamics and performance criteria provided by the user.
- **Task Analysis**: Provide a brief analysis of the system and justify the selection of the task-specific agent.
- **Agent Number**: Specify the task-specific agent number (choose from 1 to 6).
### Example of the expected JSON format:

```
{
  "Task Requirement": "[Summarize the system dynamics and
      performance criteria provided by the user]",
  "Task Analysis": "[High level task analysis]",
  "Agent": "[Task-specific agent number: 1, 2, 3, 4, 5, or 6]"
}
```

---

---

**Task-Specific Agent Prompt (First-order Stable)**

You are a control engineer expert, and your goal is to design a controller $K(s)$ for a system with transfer function $G(s)$ using loop shaping method. The loop transfer function is $L(s) = G(s)K(s)$ and here are the basic loop shaping steps:

**[Step1]** Choose a proper loop bandwidth $\omega_L$ for the given plant $G(s)$.
Note: Increasing $\omega_L$ will make the response faster, therefore smaller settling time. On the other hand, decreasing $\omega_L$ corresponds to larger settling time.

**[Step2]** Compute the proportional gain $K_p$ to set the desired loop bandwidth $\omega_L$, where $K_p = \pm 1/|G(j\omega_L)|$.

**[Step3]** Design an integral boost to increase the low frequency loop gain thus improving both tracking and disturbance rejection at low frequencies. Specifically, select $K_i(s) = (\beta_b s + \omega_L)/(s\sqrt{\beta_b^2 + 1})$ with $\beta \geq 0$. A reasonable initial choice of $\beta_b$ is $\sqrt{10}$.
Note: Decreasing beta will: (i) increase the low frequency gain and reduce the high frequency gain thus improving both tracking and noise rejection performance, and (ii) reduce the phase at loop crossover thus degrading robustness. Hence a smaller $\beta_b$ should only be used if the loop can tolerate the reduced phase. On the other hand, increasing beta will increase the phase margin.

Thus the final controller is then: $K = K_p K_i(s)$. There are two key design parameters for loop shaping: $\omega_L$ and $\beta_b$. Your goal is to find a proper combination of these two parameters such that the designed controller achieves satisfactory performance, such as phase margin and settling time requirements.
You will also be provided by a list of your history design and the corresponding performance if there is any. And you should improve your previous design based on the user request.
Note: If you could not see an improvement within 3 rounds, to make the tuning process more efficient, please be more aggressive and try to increase design step based on the previous designs.

In the above, we have provide a guideline to design a loop-shaping controller for the first-order stable systems along with the parameter tuning instructions (highlighted in purple) to help LLM agent perform tuning task.

---

**Task-Specific Agent Response Instruction**

## Response Instructions:
Please provide the controller design to the given plant G(s). Your response should strictly adhere to the following JSON format, which includes two keys: 'design' and 'parameter'. The 'design' key can contain design steps and rationale about the parameters choice or the reason to update specific parameter based on the previous design and performance, and the 'parameter' key should ONLY provide a list of numerical values of the chosen parameters.

### Example of the expected JSON format:

```
{
    "design": "[Detailed design steps and rationale behind
        parameters choice]",
    "parameter": "[List of Parameters]"
}
```

---

## E.2 FULL PROMPTS FOR BASELINES

In this section, we present the full prompt used to measure baseline approaches. We evaluate the designed controller by requesting specific output format so that we can extract the controller parameters.

---

**Zero-shot prompt CoT**

You are a control engineer expert, and your goal is to design a controller $K(s)$ for a system with transfer function $G(s)$. Please design the controller for the following system:

$$G(s) = \frac{19.95}{s + 0.4}.$$

Design the controller to meet the following specifications:
Phase margin greater or equal 71.54 degrees,
Settling time greater or equal 0.0048 sec,
Settling time should also be less or equal to 3.7264 sec,
Steady state error less or equal 0.0001.

**Please perform the design process step by step.**

---

Zero-shot prompt is almost identical to the zero-shot CoT without asking the LLM to perform the design process step by step explicitly.

---

**Few-shot CoT prompt**

You are a control engineer expert, and your goal is to design a controller $K(s)$ for a system with transfer function $G(s)$.

To help you complete the task, we provide the following demonstration example:

## Example 1
Design a controller for the first-order system $G(s) = 7/(s + 3)$.
Design the controller to meet the following specifications:
Phase margin greater or equal 90 degrees,
Settling time greater or equal 3 sec,
Settling time should also be less or equal to 6 sec,
Steady state error less or equal 0.0001.
A successful controller design is $K(s) = (1.917s + 1.818)/3.317s$.

**Let's design the controller step by step:** The plant G(s) has a pole at $s = 3\,\mathrm{rad/\,sec}$. To meet the specified requirements of phase margin, settling time, and steady state error. We first select a crossover frequency $\omega_L = 3\,\mathrm{rad/\,sec}$ to achieve desired response. Then the controller gain is selected as $K_g = 1/|G(j\omega_L)| = 0.6$. Then we choose the integral boost $K_b(s) = (\beta_b s + \omega_b)/\sqrt{\beta_b^2 + 1}s$ with $\beta_b = \sqrt{10}$ and $\omega_b = \omega_L = 15$. Therefore, the final controller $K(s) = K_g K_b(s) = (1.917s + 1.818)/3.317s$.

## Example 2 ...

Now please design the controller for the following system:

$$G(s) = \frac{19.95}{s + 0.4}.$$

Design the controller to meet the following specifications:
Phase margin greater or equal 71.54 degrees,
Settling time greater or equal 0.0048 sec,
Settling time should also be less or equal to 3.7264 sec,
Steady state error less or equal 0.0001.

**Please perform the design process step by step.**

---

Few-shot prompt is almost identical to the few-shot CoT without detailed reasoning steps and asking the LLM to perform the design process step by step explicitly.

# F   MORE DETAILS ON CONTROLEVAL

In this section, we present more details on the construction of ControlEval. We first discuss the dynamical models considered in ControlEval in Section F.1, then the associated performance criteria is discussed in Section F.2. Finally, we discuss the dataset construction in Section F.3.

## F.1   DYNAMICAL SYSTEM MODELS

This section provides a detailed overview of the various types of dynamical system models included in ControlEval, such as stable and unstable first-order systems, stable and unstable second-order systems, first-order systems with time delay, and higher-order systems. As mentioned in the main paper, a general transfer function for a dynamical system can be expressed as:

$$G(s) = \frac{Y(s)}{U(s)} = \frac{b_m s^m + b_{m-1} s^{m-1} + \cdots + b_1 s + b_0}{a_n s^n + a_{n-1} s^{n-1} + \cdots + a_1 s + a_0}, \tag{5}$$

where $Y(s)$ and $U(s)$ are output and input, $s$ is the complex frequency variable in Laplace domain. The system is considered stable if all roots of the characteristic equation $U(s) = 0$ have negative real parts. Depending on the form of $U(s)$ and $Y(s)$, we classify the dynamical system models as follows:

**First-order systems.**   A first-order system is characterized by a first-order polynomial in $s$ for $U(s)$. The corresponding transfer function is

$$G(s) = \frac{K}{\tau s + 1},$$

where $K$ is a constant gain, and $\tau$ is the time constant of the first-order system.

**Second-order systems.**   Stable second-order systems can be expressed as:

$$G(s) = \frac{a}{s^2 + 2\zeta\omega_n s + \omega_n^2}, \tag{6}$$

where $\omega_n$ is the natural frequency and $\zeta$ is the damping ratio.

**First-order systems with time delay.**   First-order systems with time delay incorporate a delay parameter $\theta$ into the transfer function:

$$G(s) = \frac{K e^{-\theta s}}{\tau s + 1},$$

where $\theta > 0$ is the time delay parameter, $K$ is the system gain, and $\tau$ is the system time constant. The presence of $e^{-\theta s}$ in the numerator introduces phase lag, which can significantly impact the system's stability and response.

**Higher-order system.**   Higher-order systems refer to systems where the order of $U(s)$ is three or greater. Higher-order systems can exhibit more complex dynamics, such as multiple resonant peaks or oscillatory behavior, making their stability analysis and control design more challenging.

## F.2   PERFORMANCE CRITERIA

In linear control system design, key performance criteria include stability, phase margin, and settling time, which are essential for ensuring both the functionality and robustness of the system.

**Stability** is a fundamental requirement in control systems. Formally, as discussed above, a LTI system is stable if all the poles of its transfer function lie in the left half of the complex s-plane, meaning their real parts are negative. If any pole has a real part greater than or equal to zero, the system is either marginally stable or unstable. Stability ensures that the system's output will return to its equilibrium state after a disturbance, without unbounded oscillations or divergence.

**Phase margin** $\phi_m$ is a measure of how close a system is to instability. It is defined as the difference between the phase angle of the system's open-loop transfer function $L(j\omega) = G(j\omega)C(j\omega)$ and $-180°$ at the gain crossover frequency $\omega_{gc}$, which is the frequency where the magnitude of the open-loop transfer function is equal to 1 (or 0 dB). Formally, the phase margin PM is given by:

$$\text{PM} = 180° + \arg(L(j\omega_{gc})). \tag{7}$$

A positive phase margin indicates a stable system, with typical design criteria recommending phase margins between $45°$ and $90°$ for adequate robustness. Phase margin provides insight into how much additional phase lag the system can tolerate before becoming unstable.

**Settling time** $T_s$ is a critical metric for evaluating the transient response of a control system. It is defined as the time required for the system's output to remain within a specified percentage (typically 2% or 5%) of its final steady-state value following a step input. Depending on the specific dynamical system, the required settling time can vary significantly—ranging from fast to slow responses—based on factors such as system type, stability requirements, and the presence of unmodeled dynamics. To account for these variations, we consider three distinct response modes for stable first-order and second-order systems in ControlEval.

### F.3 ControlEval Generation Details

In this section, we explain the details on the construction of ControlEval.

#### F.3.1 Requirements for Different Type of Dynamical Models.

**First-order Stable Systems** For first-order stable systems, we sample $K \sim \mathcal{U}(0.005, 200)$, i.e., we uniform sample $K$ from range $[0.005, 200]$, and $\tau \sim \mathcal{U}(0.05, 10)$. In addition, we require the settling time to be within some range $T_s \in [t_{\min}, t_{\max}]$ for three different response modes: fast, moderate, and slow relative to the time constant $\tau$:

- Fast mode: We have $t_{\min} \sim \mathcal{U}(0, 0.001\tau)$ and $t_{\max} \sim \mathcal{U}(0.3\tau, 0.5\tau)$.
- Moderate mode: We have $t_{\min} \sim \mathcal{U}(0.1\tau, 0.5\tau)$ and $t_{\max} \sim \mathcal{U}(\tau, 5\tau)$.
- Slow mode: We have $t_{\min} \sim \mathcal{U}(5\tau, 10\tau)$ and $t_{\max} \sim \mathcal{U}(20\tau, 30\tau)$.

The minimum required phase margin is also randomly seleted by $\phi_m \in [45, 90]$.

**Second-Order Stable Systems.** For second-order stable systems, we sample $\zeta \sim \mathcal{U}(0.1, 0.99)$ to consider underdamped second-order system, $\omega_n \sim \mathcal{U}(0.1, 5)$, and $a \sim \mathcal{U}(0.1, 20)$. In addition, we require the settling time to be within some range $T_s \in [t_{\min}, t_{\max}]$ for three different response modes: fast, moderate, and slow relative to the time constant $\tau \approx \frac{4}{\zeta\omega_n}$:

- Fast mode: We have $t_{\min} \sim \mathcal{U}(0, 0.005\tau)$ and $t_{\max} \sim \mathcal{U}(\tau, 1.5\tau)$.
- Moderate mode: We have $t_{\min} \sim \mathcal{U}(2\tau, 2.5\tau)$ and $t_{\max} \sim \mathcal{U}(3\tau, 4\tau)$.
- Slow mode: We have $t_{\min} \sim \mathcal{U}(4\tau, 5\tau)$ and $t_{\max} \sim \mathcal{U}(6\tau, 10\tau)$.

The minimum required phase margin is also randomly seleted by $\phi_m \in [45, 65]$.

**Second-Order Unstable Systems.** For the second-order unstable systems, half of the dataset is generated using the following transfer function structure:

$$G(s) = \frac{a}{s^2 - 2\zeta\omega_n s + \omega_n^2}, \tag{8}$$

where the damping ratio $\zeta$ is sampled from a uniform distribution $\zeta \sim \mathcal{U}(0.1, 0.99)$, ensuring the system has two unstable poles. The natural frequency $\omega_n$ is sampled from $\omega_n \sim \mathcal{U}(0.1, 5)$, and the gain $a$ is drawn from $a \sim \mathcal{U}(0.1, 20)$. Additionally, the settling time $T_s$ is constrained to lie within the range $T_s \in [t_{\min}, t_{\max}]$, where $t_{\min} \sim \mathcal{U}(0, 0.05\tau)$ and $t_{\max} \sim \mathcal{U}(\tau, 1.5\tau)$, with the time constant $\tau \approx \frac{4}{\zeta\omega_n}$. The minimum required phase margin $\phi_m$ is randomly selected from $\phi_m \sim \mathcal{U}(45, 65)°$.

The second half of the dataset is designed using the following transfer function structure:

$$G(s) = \frac{a}{(\tau_1 s + 1)(\tau_2 s + 1)},\tag{9}$$

where the gain $a$ is sampled from $a \sim \mathcal{U}(-2000, -0.00025)$, and the time constants $\tau_1$ and $\tau_2$ are drawn from $\tau_1 \sim \mathcal{U}(0.05, 10)$ and $\tau_2 \sim \mathcal{U}(-10, -0.05)$, respectively. The time constant $\tau$ is approximated as $\tau \approx 3\min\{\tau_1, |\tau_2|\}$. As with the first half of the dataset, the settling time $T_s$ is required to be within the range $T_s \in [t_{\min}, t_{\max}]$, with $t_{\min} \sim \mathcal{U}(0, 0.05\tau)$ and $t_{\max} \sim \mathcal{U}(\tau, 1.5\tau)$.

The reason for using these two different structures is to ensure that half of the dataset consists of systems with one unstable pole, while the other half contains systems with two unstable poles, providing a balanced variety of unstable system dynamics.

**First-Order Systems with Time Delay.** For first-order system with a time delay $\theta$, we choose the delay parameter randomly as $\theta \sim \mathcal{U}(\theta_{\min}, \theta_{\max})$, where $\theta_{\min} = 0.1\tau$ and $\theta_{\max} = 0.2\tau$. For the settling time requirements, we require $T_s \in [t_{\min}, t_{\max}]$ with $t_{\min} \sim \mathcal{U}(4\tau, 5\tau)$ and $t_{\max} \sim \mathcal{U}(40\tau, 50\tau)$. The minimum required phase margin is also randomly selected by $\phi_m \in [45, 65]$.

**Higher-order Systems.** For higher-order systems, there is no standardized method to automatically generate both the dynamical system and its corresponding performance requirements simultaneously. To address this, we have manually designed 50 higher-order systems along with their performance criteria, ensuring that the specified requirements are indeed achievable. Among these samples, 25 are stable higher-order systems, while the remaining 25 are unstable. Based on feedback from human experts, designing a controller for higher-order systems is challenging, requires multiple rounds of parameter tuning even for human. Nevertheless, these higher-order tasks present unique challenges for ControlAgent, and our results demonstrate promising performance in addressing these complex scenarios.

### F.3.2 SAMPLE CODE FOR GENERATING CONTROLEVAL

The following is a sample code snap for generating first-order systems:

```python
def generate_first_order_system_dataset(num_samples):
    dataset = []

    for i in range(num_samples):
        K = random.uniform(0.1, 20)
        B = random.uniform(0.1, 20)
        tau = 3/B

        phase_margin_min = random.uniform(45, 90)
        settling_time_min_fast = random.uniform(0, 0.001 * tau)
        settling_time_max_fast = random.uniform(0.3* tau, 0.5 * tau)
        settling_time_min_moderate = random.uniform(0.1*tau, 0.5*tau)
        settling_time_max_moderate = random.uniform(tau, 5 * tau)
        settling_time_min_slow = random.uniform(5 * tau, 10 * tau)
        settling_time_max_slow = random.uniform(20 * tau, 30 * tau)
        steadystate_error_max = 0.0001
        metadata = "First order system with different response speed
                                        requirements."

        system_data = {
            "id": i,
            "num": [K],
            "den": [1, B],
            "phase_margin_min": phase_margin_min,
            "settling_time_min_fast": settling_time_min_fast,
            "settling_time_max_fast": settling_time_max_fast,
            "settling_time_min_moderate": settling_time_min_moderate,
            "settling_time_max_moderate": settling_time_max_moderate,
            "settling_time_min_slow": settling_time_min_slow,
            "settling_time_max_slow": settling_time_max_slow,
            "steadystate_error_max": steadystate_error_max,
            "metadata": metadata
        }

        dataset.append(system_data)

    return dataset
```

The following is a sample code snap for generating second-order unstable systems:

```python
def generate_unstable_second_order_system_dataset(num_samples):
    dataset = []

    for i in range(num_samples):
        if i % 2 == 0:

            zeta = random.uniform(0.1, 0.99)
            omega = random.uniform(0.1, 5)
            A = random.uniform(0.1,20)
            sT = 4/(omega*zeta)

            phase_margin_min = random.uniform(45, 65)
            settling_time_min = random.uniform(0, 0.05 * sT)
            settling_time_max = random.uniform(sT, 1.5 * sT)
            overshoot_max = random.uniform(5, 20)
            steadystate_error_max = 0.0001
            metadata = "Second order unstable system with different
                                            response speed
                                            requirements."

            system_data = {
                "id": i,
                "num": [A],
                "den": [1, -2*zeta*omega, omega*omega],
                "phase_margin_min": phase_margin_min,
                "settling_time_min": settling_time_min,
                "settling_time_max": settling_time_max,
                "steadystate_error_max": steadystate_error_max,
                "metadata": metadata
            }

        else:

            A = random.uniform(0.1,20)
            B = random.uniform(0.1, 20)
            C = random.uniform(0,-20)
            sT = 3/min(B,abs(C))

            phase_margin_min = random.uniform(45, 65)
            settling_time_min = random.uniform(0, 0.05 * sT)
            settling_time_max = random.uniform(sT, 1.5 * sT)
            overshoot_max = random.uniform(5, 20)
            steadystate_error_max = 0.0001
            metadata = "Second order unstable system with different
                                            response speed
                                            requirements."

            system_data = {
                "id": i,
                "num": [A],
                "den": [1, B+C, B*C],
                "phase_margin_min": phase_margin_min,
                "settling_time_min": settling_time_min_fast,
                "settling_time_max": settling_time_max_fast,
                "steadystate_error_max": steadystate_error_max,
                "metadata": metadata
            }

        dataset.append(system_data)

    return dataset
```

# G  A DEMO OF THE DESIGN PROCESS

In this section, we present a step-by-step demo of the ControlAgent design process for an illustrative control task. The goal is to design a controller for a given dynamical system that satisfies specific performance requirements, including settling time and phase margin. The design process begins by providing ControlAgent with the initial system models and desired performance criteria. The central agent first performs high-level task analysis and assign the task to a task-specific LLM agent, who then iteratively refines its controller design. At each iteration, ControlAgent evaluates the proposed design against the desired criteria and, if necessary, adjusts the controller parameters based on the observed performance. The iterative refinement continues until a design that meets all requirements is achieved or the maximum iteration number is achieved. Table 15 shows a breakdown of LLM agents in ControlAgent.

| LLM Agents | Role |
|---|---|
| Central Agent | High-level task analysis and task distribution |
| Task-Specific Agent 1 | Controller design for first-order stable system |
| Task-Specific Agent 2 | Controller design for first-order unstable system |
| Task-Specific Agent 3 | Controller design for second-order stable system |
| Task-Specific Agent 4 | Controller design for second-order unstable system |
| Task-Specific Agent 5 | Controller design for first-order system with time delay |
| Task-Specific Agent 6 | Controller design for higher-order system |

Table 15: Overview of the LLM agents and their roles in ControlAgent.

---

**User Input**

Please design the controller for the following system:

$$G(s) = \frac{19.95}{s + 0.3897}.$$

Design the controller to meet the following specifications:

- The system should be stable and steady state error less or equal 0.0001.
- Phase margin greater or equal 71.542 degrees,
- Settling time greater or equal 0.005 sec,
- Settling time should also be less or equal to 3.726 sec.

---

**Central Agent Output**

**Task Requirement**: The system provided is a first-order stable system represented by the transfer function $\frac{19.95}{(s+0.3897)}$. The control design must ensure a phase margin of at least 71.542 degrees, a settling time between 0.005 sec and 3.726 sec, and a steady state error of less than or equal to 0.0001.

**Task Analysis**: The given transfer function, $\frac{19.95}{(s+0.3897)}$, clearly indicates a first-order system due to the presence of a single $s$ term in the denominator. The positive coefficient of s (0.3897) in the denominator suggests that the system is stable. The requirements for phase margin, settling time, and steady state error are specific performance criteria that need to be met through appropriate controller design. Given the nature of the system as first-order and stable, Agent 1, who specializes in first-order stable systems, is the most suitable choice for designing the required controller.

**Agent**: Agent 1 for first-order stable systems controller design.

In the outputs of the central agent, one can see that the central agent is able to perform an accurate high-level task analysis and assign the task to the corresponding task-specific agent accurately.

---

**Task-Specific Agent Output**

**Iteration 1:**
To design the controller $K(s)$ for the given plant $G(s) = 19.95/(s+0.3897)$, we will follow the loop shaping method. First, we need to choose a proper loop bandwidth $\omega_L$. Given the settling time requirements, we aim for a moderate $\omega_L$ to balance between fast response and robustness. Let's start with $\omega_L = 1\,rad/s$, which is a reasonable initial choice for a system with a pole at $s = -0.3897$. Next, we calculate the proportional gain $K_p$ to set the desired loop bandwidth. At $\omega_L = 1$, $|G(j1)| = 19.95/\sqrt{1^2 + 0.3897^2} = 19.95/1.073 = 18.6$. Therefore, $K_p = 1/18.6 = 0.0538$. For the integral boost, we select $\beta_b = \sqrt{10} \approx 3.1623$ to start with, which provides a good balance between low-frequency gain and phase margin. The integral boost $K_i(s)$ is then $(3.1623s + 1)/(s\sqrt{3.1623^2 + 1})$. The final controller is $K(s) = K_p K_i(s)$. This initial design aims to meet the phase margin and settling time requirements while ensuring a low steady-state error.

---

In the first iteration, Task-specific agent 1 is able to follow the design instruction and come up with a reasonable initial controller.

---

**Task-Specific Agent Output**

**Iteration 2:**
In the first design, the settling time was too long, exceeding the maximum requirement of 3.726 seconds. To reduce the settling time, we need to increase the loop bandwidth $\omega_L$. This will make the system response faster. Let's increase $\omega_L$ to 2, which should help in reducing the settling time. We will keep $\beta_b$ at 3.1623 initially to maintain a good balance between low-frequency gain and phase margin. After adjusting $\omega_L$, we will calculate the proportional gain $K_p$ to set the desired loop bandwidth. The integral boost $K_i$ will be designed using the updated $\omega_L$ and the initial $\beta_b$ value.

---

For iteration 2, the agent is able to identifying the issues in the previous design and come up with a correct tuning direction, i.e., increasing $\omega_L$ to reduce the settling time.

---

**Task-Specific Agent Output**

**Iteration 3:**
In the previous designs, the settling time was too long, exceeding the maximum requirement of 3.726 seconds. To reduce the settling time, we need to increase the loop bandwidth $\omega_L$. In Design 2, $\omega_L$ was set to 2, resulting in a settling time of approximately 3.993 seconds. To further reduce the settling time, I will increase $\omega_L$ to 3. This should help bring the settling time within the desired range. The $\beta_b$ parameter, which affects the phase margin, was previously set to 3.1623. Since the phase margin in Design 2 was 83.4768 degrees, which is well above the required 71.5421 degrees, we can afford to slightly reduce $\beta_b$ to 2.5 to maintain a good balance between phase margin and settling time. This adjustment should help achieve the desired performance specifications.

---

The agent further increases $\omega_L$ to reduce the settling time. Additionally, upon evaluating the previous design, the agent observes that it adequately satisfies the phase margin. Consequently, it can fine-tune $\beta_b$ to maintain an optimal balance between system performance and robustness. After iteration 3, the task-specific agent produces a successful controller design, which is verified by the Python computation agent. The final controller is returned to the user as the output of ControlAgent.

