# OpenReview forum: "ControlAgent: Automating Control System Design via Novel Integration of LLM Agents and Domain Expertise"
_ICLR.cc/2025/Conference — Submitted to ICLR 2025_

### Official Review · Reviewer_W6Fy · 2024-10-18

**Soundness:** 1
**Presentation:** 2
**Contribution:** 1
**Rating:** 6
**Confidence:** 3

**Summary:**

This paper describes a composite LLM-based system for control tasks which attempts to design controllers, represented as Python code, for control problems with specific requirements, namely stability, phase margin, and settling time.

While this paper is decently presented and seems to achieve decent results, I am uncertain about recommending it for ICLR. Primarily, the paper seems highly domain-specific and engineering-focused, rather than more general cutting-edge academic research. Still, it is a good engineering system. Secondly, I am uncertain about the evaluation.

The proposed method is essentially a domain-specific application of LLM-modulo, e.g. an interative prompt with a verifier and critiques [1].

[1] Kambhampati, S., Valmeekam, K., Guan, L., Verma, M., Stechly, K., Bhambri, S., ... & Murthy, A. B. Position: LLMs Can’t Plan, But Can Help Planning in LLM-Modulo Frameworks. In Forty-first International Conference on Machine Learning.

**Strengths:**

This paper addresses the issue of designing controllers using LLMs, in particular with specific stability, phase margin, and settling times.

The overall system runs in a loop where a the designed controller is run and the system provides feedback based on a history of designs and how well they performed.

**Weaknesses:**

It seems guarantees would be desirable when working with control systems, and I assume the problem requirements are meant to be guarantees. However, I feel the paper would be made a lot stronger by discussing guarantees at length.

The evaluation methods seem like they could be improved, in particular I would like the authors to clarify about "a system is considered successfully designed if at least one of the multiple independent trials results in a successful design". It seems this would greatly skew the statistics, since failures are being filtered out. I also don't see reporting of how many samples are taken to achieve the reported success rates.

Given the unpredictable and error-prone nature of LLMs, I am skeptical that the overall system can work without a human in the loop or method for filtering correct answers. Also, it seems like intermediate mistakes in generation (e.g. a hallucinated constant) would collapse the entire system, so I would expect it to be rather fragile. To the extent that the proposed method works, I am curious what the authors attribute it to?

While the method is interesting, it seems to be an incomplete solution to a highly domain-specific problem, so I'm unsure about the larger impact of the work, e.g. the paper doesn't give much insight into designing general LLM-based systems.

**Questions:**

How much sampling is done of LLM-generated designs? e.g. is the budget 10 designs?

---

> ### Author Response · Authors · 2024-11-25
> **Author Response, Part 1**
>
> We appreciate the reviewers' valuable comments and have provided our responses below. We realize that the reviewer has misunderstood our contributions, and provide extensive clarifications to address this. In addition, based on the reviewer’s suggestions, we have also **added new results** on the more robust evaluations with pass@k metric and sample numbers needed for each control problem type. We hope these responses address your concerns, and we hope you can reevaluate our paper based on our new results and we also welcome any further feedback.
>
> **Q1. While this paper is decently presented and seems to achieve decent results, I am uncertain about recommending it for ICLR. Primarily, the paper seems highly domain-specific and engineering-focused, rather than more general cutting-edge academic research. Still, it is a good engineering system.**
>
> Thanks for the positive feedback on the presentation and for acknowledging our results. We really believe that our research is cutting-edge academic research that is particularly relevant to ICLR. As stated by the ICLR 2025 Reviewer Guide, a paper brings value to the ICLR community when it convincingly demonstrates new, relevant, impactful knowledge. A key academic contribution of our paper is the novel integration of two major disciplines - control engineering and large language models. Control engineering is not merely a domain-specific application, but rather a fundamental discipline that underpins modern technology and civilization (from aerospace and robotics to power systems and autonomous vehicles) and has rich mathematical foundations and design principles developed over decades. Our paper presents the first approach to systematically bridging these two disciplines by showing how to encode complex control engineering principles into LLM agents and combine control-oriented computation tools with LLM reasoning to enable ControlAgent to automatically navigate the fundamental performance-robustness tradeoffs in control design. This is beyond a naive integration of control and LLMs. Our key contributions include:
>
> 1. Iterative Refinement Mechanism: Mimicking control engineers' decision-making processes to navigate performance-robustness trade-offs.
> 2. Structured Memory Management: Allowing LLMs to learn from prior designs while respecting context limitations.
> 3. Multi-Agent Architecture: Enabling specialized LLM agents to collaborate effectively, ensuring technical correctness and efficiency.
>
> Just as the integration of deep learning with physics, chemistry, or biology has led to significant advances in both AI and these respective fields (e.g., AlphaFold), our work shows how the marriage of LLMs with control engineering can push the boundaries of both disciplines. The insights gained from enabling LLMs to understand and apply control engineering principles could inform how we approach the integration of LLMs with other engineering disciplines.
>
> In addition, the ICLR community encourages applications to various domains such as robotics, autonomy, and planning, etc. We see no reason why engineering applications should be excluded from consideration. In fact, prior ICLR has accepted papers focused on LLM applications in application domains such as operations research and geospatial tasks [1,2].
>
> [1] Xiao, Z., Zhang, D., Wu, Y., Xu, L., Wang, Y.J., Han, X., Fu, X., Zhong, T., Zeng, J., Song, M. and Chen, G., 2023. Chain-of-Experts: When LLMs Meet Complex Operations Research Problems. ICLR.
>
> [2] Manvi, R., Khanna, S., Mai, G., Burke, M., Lobell, D. and Ermon, S., 2023. Geollm: Extracting geospatial knowledge from large language models. ICLR.
>
> Our research addresses core challenges in AI/ML research, particularly how to enable LLMs to perform reliable technical reasoning and design in disciplines requiring deep theoretical understanding and sharp engineering insights. The solutions we develop - including the structured integration of engineering knowledge, iterative refinement mechanisms, and multi-agent coordination - represent important advances in LLMs for engineering.
>
> Our empirical validation via ControlEval and the new added results on real-world applications (see our general response, part 2), systematically demonstrates ControlAgent’s effectiveness compared to traditional tools and baseline LLM approaches. The strong performance highlights the value of integrating rigorous engineering principles with LLM reasoning, showcasing a practical path toward leveraging AI for real-world, complex problem-solving. We also believe that our work offers useful insights beyond control by demonstrating how LLMs can be leveraged to solve complex, real-world engineering problems.
>
> To sum up, the fundamental nature of both control engineering and LLMs, combined with the technical depth and novelty of our approach, makes this work relevant to ICLR, which has a strong tradition of publishing papers that advance ML methodology through innovative integration with other disciplines.

---

> ### Author Response · Authors · 2024-11-25
> **Author Response, Part 2**
>
> **Q2. The proposed method is essentially a domain-specific application of LLM-modulo, e.g. an interative prompt with a verifier and critiques [1].**
>
> We thank the reviewer for bringing the LLM-modulo framework [1] to our attention. We have included this reference in our updated paper. While we agree that there are similarities between ControlAgent and LLM-modulo frameworks, we would like to clarify the key differences and highlight the unique aspects of our approach. Specifically, our ControlAgent addresses the unique challenges specific to engineering design. The verifier and critique framework from LLM-modulo does not tell us how to re-design the engineering solutions based on all the past outcomes from the verifier and critiques. Our ControlAgent combines domain expertise with LLM reasoning to perform such iterative design processes that mimic how human control engineers iterate control system design. In addition, ControlAgent tackles design tasks for a wide range of dynamic systems from stable first-order systems to unstable higher-order systems, creating a fully automated design pipeline (we emphasize that there is no human intervention). By integrating domain-specific instructions, implementing an iterative design process with accurate feedback, conducting a robust, deterministic evaluation, and introducing the ControlEval evaluation dataset, ControlAgent sets itself apart from other approaches and demonstrates its capability in handling complex, real-world engineering tasks.
>
> [1] Kambhampati, S., Valmeekam, K., Guan, L., Verma, M., Stechly, K., Bhambri, S., ... & Murthy, A. B. Position: LLMs Can’t Plan, But Can Help Planning in LLM-Modulo Frameworks. In Forty-first International Conference on Machine Learning.
>
> **Q3. It seems guarantees would be desirable when working with control systems, and I assume the problem requirements are meant to be guarantees. However, I feel the paper would be made a lot stronger by discussing guarantees at length.**
>
> We thank the reviewer for this suggestion. We have added more detailed discussions on the guarantees in Section B.3, B.4, and B.5. Yes, the designed controllers from ControlAgent have strong performance guarantees in term of settling time and strong robustness guarantees in term of phase margins. As commented by Reviewer AbjT, the iterative design process of ControlAgent is noteworthy for its theoretical soundness. Our ControlAgent framework is the first LLM agent framework in automating the process of designing controllers that can rigorously meet pre-specified requirements on performance (settling time) and robustness (phase margin) and effectively navigate the fundamental performance/robustness trade-off in classic control design. ControlAgent uses a Python computation sub-agent to automatically and exactly evaluate the stability, settling time, and phase margin of the designed controllers, and put such exact information into a memory module such that the LLM sub-agents exactly know whether the current designed controller meets the user-specified performance/robustness requirements or not. The Python computation sub-agent is integrated as a module of ControlAgent and works in a fully automated manner to ensure the ControlAgent always know the true settling time and phase margin of its own designed controllers. Whenever ControlAgent gives a successful design, it has to pass the verification of the Python computation agent such that the performance guarantees on settling time and the robustness guarantees on phase margins are both ensured.

---

> ### Author Response · Authors · 2024-11-25
> **Author's Response, Part 3**
>
> **Q4. Clarify about "a system is considered successfully designed if at least one of the multiple independent trials results in a successful design". It seems this would greatly skew the statistics, since failures are being filtered out.**
>
> We thank the reviewer for the constructive comment. We have added a new experiment to evaluate a more robust evaluation with pass@k metric in Section D.4.2 in our revised paper. However, we would like to mention that our main evaluation metric is averaged successful rate (ASR), which is computed as the fraction of successful design among multiple independent trials. Formal definitions of ASR can be found in Section D.2 in our revision. ASR is robust and should not skew the statistics. From Table 1 (we reported ASR not AgSR) in our main paper, we have shown the superior performance of ControlAgent to the baseline methods. AgSR is designed in an ensemble manner, similar to metrics used in previous works such as [1]. This approach is meaningful in practical applications, particularly in control system design, where it is often feasible to use the Python Computation Agent to automatically filter out failed designs and pick the best design solution.
>
> We agree that AgSR could potentially introduce high variance. To address this, we employed the advanced pass@k metric as presented in [2], which is designed to be unbiased and reduces variance. We computed pass@k with k = {1, 3, 5}  and n = 5, as detailed below:
>
> |        | 1st f | 1st m | 1st s | 2nd f | 2nd m | 2nd s | 1st un | 2nd un | 1 wd | high |
> |--------|-------|-------|-------|-------|-------|-------|--------|--------|------|------|
> | pass@1 | 1.00  | 1.00  | 1.00  | 1.00  | 1.00  | 1.00  | 0.98   | 0.91   | 0.97 | 0.82 |
> | pass@3 | 1.00  | 1.00  | 1.00  | 1.00  | 1.00  | 1.00  | 1.00   | 0.99   | 1.00 | 0.95 |
> | pass@5 | 1.00  | 1.00  | 1.00  | 1.00  | 1.00  | 1.00  | 1.00   | 1.00   | 1.00 | 0.96 |
>
> It can be seen that ControlAgent is able to achieve high pass@k rates, even for complex tasks, highlights its robustness and effectiveness in handling diverse control problems.
>
> [1] Kulal, S., Pasupat, P., Chandra, K., Lee, M., Padon, O., Aiken, A. and Liang, P.S., 2019. Spoc: Search-based pseudocode to code. NeurIPS.
>
> [2] Chen, M., Tworek, J., Jun, H., Yuan, Q., Pinto, H.P.D.O., Kaplan, J., Edwards, H., Burda, Y., Joseph, N., Brockman, G. and Ray, A., 2021. Evaluating large language models trained on code.
>
> **Q5. Given the unpredictable and error-prone nature of LLMs, I am skeptical that the overall system can work without a human in the loop or method for filtering correct answers. I would expect it to be rather fragile.**
>
>  We emphasize that ControlAgent is fully automated and there is no human in the loop. We even provide a real DC motor example in C.2 to showcase that ControlAgent can work for real problems without human in the loop. Please see our general response. More explanations are provided to explain why our approach is not fragile and can work reliably.
>
> **Constrained Reasoning Space for LLMs**: We acknowledge the reviewer’s skepticism about the unpredictability of LLMs. However, in our framework, the LLM’s reasoning space is constrained by the problem context and task-specific requirements. By distilling control knowledge in loop shaping as system prompts for LLM agents, the likelihood of generating irrelevant or incorrect content is reduced.
>
> **Robust Python Computation Agent**: Our Python Computation Agent is exact and deterministic, free from any generative hallucinations. This agent handles all numerical computations and logical verifications, ensuring precise and reliable outcomes. The determination of whether a design is successful or not is made solely by the Computation Agent, not the LLM. Thus, this step is exact, leaving no room for hallucinations to affect the overall system.
>
> **Accurate Feedback Mechanism**:  The feedback module in our framework is designed to be exact and accurate. Intermediate designs generated by the LLM are passed to the Computation Agent for validation, which also rigorously checks the current design against task-specific requirements and provides precise feedback to guide subsequent LLM designs. By continuously validating and refining the LLM’s designs, we mitigate potential hallucinations and ensure robust system performance.
>
> **Iterative Design Process**: Although the risk of hallucination is inherent in LLMs, our framework additionally mitigates this issue through a multi-step reflection process. We employ an iterative design and verification process, where the LLM reflects on its previous designs and adjusts accordingly, based on exact feedback. This iterative approach also enhances robustness/reliability.
>
> The superior performance of ControlAgent on ControlEval (and extra applications in Appendix C) supports our claim that ControlAgent has addressed the potential issues from LLM hallucinations and can operate reliably without requiring a human in the loop.

---

> ### Author Response · Authors · 2024-11-25
> **Author's Response, Part 4**
>
> **Q6. While the method is interesting, it seems to be an incomplete solution to a highly domain-specific problem, so I'm unsure about the larger impact of the work, e.g. the paper doesn't give much insight into designing general LLM-based systems.**
>
> We thank the reviewer for the feedback and appreciate the recognition of our method as an interesting approach. We respectfully disagree with the assessment that our framework is limited to a highly domain-specific problem. On the contrary, we believe that ControlAgent offers a versatile and generalizable solution that can be adapted to a wide range of engineering domains and applications.
>
> **ControlAgent can solve many real-world problems**: Control engineering is not merely a domain-specific application, but rather a fundamental discipline that underpins modern technology and civilization, from aerospace and robotics to power systems and autonomous vehicles. Our ControlAgent can be readily adapted to the control of real-life applications including flight control, DC motor, high speed train control, hard disk drive control, etc. We have added a new application category in ControlEval to demonstrate the utility of ControlAgent in such applications. More details can be found in Appendix Section C in the revised paper.
>
> **General Framework Beyond a Specific Domain and Insight into LLM-based System Design**: While our primary evaluation focuses on control design, the core principles of our framework,  design, evaluation, feedback, and iterative improvement, are common features across many engineering domains.  We believe that our work not only provides a concrete case study on how to extend LLM capabilities beyond standard reasoning tasks into real-world engineering design scenarios, but also can serve as a blueprint for future AI systems targeting a wide range of complex engineering design problems. Our paper brings the following key insights into designing general LLM agent systems for serving as AI engineers.
>
> 1. Our study sheds light on the importance of using domain expertise to address context window limitations via selectively storing and providing only the essential historical information to the LLMs based on the domain expertise in the specific targeted engineering field. For ControlAgent, the memory module only retains key data such as design parameters, performance metrics, and feedback from previous iterations, while excluding unnecessary details from the LLMs’ responses. This ensures that the memory buffer remains compact and efficient. Such insights can be potentially useful for developing LLM agent systems to target other engineering design problems such as circuit design and structure design.
>
> 2. Our paper also brings general insights on how to build LLM agents to mimic human engineers. Specifically, one has to combine the design recipes from human engineers with the LLM agents.
>
> 3. Our paper highlights the importance of integrating an exact evaluation agent (in ControlAgent, it is the Python Computation Agent which gives the exact evaluations of the performance and robustness of the designed controllers) for building LLM agents in the context of engineering design.
>
> To summarize, we believe that it is fair to claim that our ControlAgent paper does bring important insights into designing general LLM agent systems for serving as AI engineers.
>
> Regarding the relevance of our paper to the ICLR community, please also see (Author Response, Part 1).

---

> ### Author Response · Authors · 2024-11-25
> **Author's Response, Part 5**
>
> **Q7. I also don't see reporting of how many samples are taken to achieve the reported success rates. How much sampling is done of LLM-generated designs? e.g. is the budget 10 designs?**
>
> For each problem in ControlEval, ControlAgent is run on five independent trials. For each trial, ControlAgent iteratively refines the controller parameters in a feedback manner with multiple conversation rounds (each conservation round requires calling the LLM API once and does not involve any human intervention). Each trial is terminated once a successful design is achieved (as determined automatically and exactly by the Python computation agent) or when ControlAgent reaches the predefined maximum number of conversation rounds per trial.  Based on the difficulty level of each problem class, the maximum number of conversation rounds per trial is set up accordingly, and the details are provided in Appendix D.1 paragraph Parameter Setup for ControlAgent (this number is set to be 10 for simpler problem class (stable 1st/2nd-order systems), and 30 for the difficult problem class (e.g. higher order)).
> For the number of samples (rounds) to achieve the reported success rates, we have reported the averaged iteration number in Table 4, where we show that for first-order stable systems, for each task, on average the ControlAgent only needs less than 3 iterations for all cases.  In the table below, we present the average number of conversation rounds (sample size) for each type of control problem:
>
> |            | 1st fast | 1st moderate | 1st slow | 2nd fast | 2nd moderate   | 2nd slow | 1st unstb | 2n unstb | 1st w dly | higher-order  |
> |----------------------|-------|-------|-------|-------|-------|-------|-------|--------|-------|-------|
> | number of rounds  | 2.744 | 1.776 | 2.193 | 2.372 | 2.640 | 3.716 | 3.900 | 5.716  | 9.908 | 9.560 |
>
>
> The results provide an average estimate of the sample efficiency of ControlAgent. As expected, the sample number needed increases as the problem type becomes more difficult. To offer a more comprehensive analysis, we also provide a breakdown of the conversation round distribution for each problem type below.
>
> | First-order-stable-fast| \# samples | [1,2) | [2,3) | [3,4) | [4,5) | [5,8) | [8,10] |         |         |
> | ---------------------------------- | ---------- | ----- | ----- | ----- | ----- | ----- | ------ | ------- | ------- |
> |                                    | **count**      | 65    | 53    | 73    | 33    | 20    | 6      |         |         |
> | **First-order-stable-moderate** | **\# samples** | **[1,2)** | **[2,3)** | **[3,4)** | **[4,5)** | **[5,7]** |        |         |         |
> |                                    | **count**      | 88    | 142   | 11    | 8     | 1     |        |         |         |
> | **First-order-stable-slow**            | **\# samples** | **[1,2)** | **[2,3)** | **[3,4]** |       |       |        |         |         |
> |                                    | **count**      | 11    | 181   | 58    |       |       |        |         |         |
> | **Second-order-stable-fast**           | **\# samples** | **[1,2)** | **[2,3)** | **[3,4)** | **[4,5)** | **[5,8)** | **[8,9]**  |         |         |
> |                                    | **count**      | 27    | 136   | 63    | 19    | 4     | 1      |         |         |
> | **Second-order-stable-moderate** | **\# samples** | **[1,2)** | **[2,3)** | **[3,4)** | **[4,5)** | **[5,8)** | **[8,10]** |         |         |
> |                                    | **count**      | 16    | 116   | 89    | 20    | 5     | 4      |         |         |
> | **Second-order-stable-slow** | **\# samples** | **[1,2)** | **[2,3)** | **[3,4)** | **[4,5)** | **[5,8)** | **[8,10]** |         |         |
> |                                    | **count**      | 2     | 80    | 76    | 38    | 21    | 33     |         |         |
> | **First-order-w-delay** | **\# samples** | **[1,2)** | **[2,3)** | **[3,4)** | **[4,5)** | **[5,8)** | **[8,10)** | **[10,15)** | **[15,20]** |
> |                                    | **count**      | 137   | 25    | 16    | 24    | 10    | 6      | 11      | 21      |
> | **First-order-unstable**               | **\# samples** | **[1,2)** | **[2,3)** | **[3,4)** | **[4,5)** | **[5,8)** | **[8,10)** | **[10,15)** | **[15,20]** |
> |                                    | **count**      | 51    | 65    | 9     | 12    | 27    | 26     | 43      | 17      |
> | **Second-order-unstable**              | **\# samples** | **[1,2)** | **[2,3)** | **[3,4)** | **[4,5)** | **[5,8)** | **[8,10)** | **[10,15)** | **[15,30]** |
> |                                    | **count**      | 0     | 6     | 8     | 4     | 68    | 60     | 64      | 40      |
> | **Higher-order**                       | **\# samples** | **[1,2)** | **[2,3)** | **[3,4)** | **[4,5)** | **[5,8)** | **[8,10)** | **[10,15)** | **[15,30]** |
> |                                    | **count**      | 17    | 23    | 46    | 30    | 22    | 12     | 30      | 70      |

---

> ### Comment · Reviewer_W6Fy · 2024-11-25
> **Response to Authors**
>
> I greatly appreciate the Author's detailed response, and apologize for my original review which, honestly, was preliminary and should've listed a lower confidence.
>
> Given the level of effort demonstrated by the authors, I will be raising my score.

---

> > ### Author Response · Authors · 2024-11-25
> >
> > We sincerely thank the reviewer for their positive support of ControlAgent! Should there be any additional comments, please feel free to share them with us, and we would be happy to discuss them further.

---

### Official Review · Reviewer_tosD · 2024-10-20

**Soundness:** 2
**Presentation:** 2
**Contribution:** 2
**Rating:** 5
**Confidence:** 3

**Summary:**

This paper introduces a new paradigm that automates control system design via novel integration of LLM agents and control-oriented domain expertise. However, the writing style is confusing, making it hard to follow their ideas. I suggest the authors improve their academic writing skills by making the abstract more precise and brief, adding the approach section, and reorganizing the corresponding method section. Moreover, I do not know what scenarios the authors implemented or simulated for the experiments. There is no background information or introduction. Generally, this paper needs to improve largely.

**Strengths:**

This paper proposes a new paradigm that automates control system design via novel integration of LLM agents and control-oriented domain expertise to bridge the the complexity and specificity in control system design.

**Weaknesses:**

The paper's writing style is confusing, making it hard to follow their ideas. I suggest the authors improve their academic writing skills by making the abstract more precise and brief, adding the approach section, and reorganizing the corresponding method section. Moreover, I do not know what scenarios the authors implemented or simulated for the experiments. There is no background information or introduction. Generally, this paper needs to improve largely.

**Questions:**

As mentioned above, I suggest the authors improve their academic writing skills and design specific application scenarios, such as robotics and transportation, to verify their framework.

I recommend several papers, as shown below, in which authors can learn how to improve academic writing skills and organize corresponding ideas from them.

1) Yang, Q., & Parasuraman, R. Bayesian strategy networks based soft actor-critic learning. ACM Transactions on Intelligent Systems and Technology (TIST).

2) H. Hamann and H. Wo ̈rn, “A framework of space–time continuous models for algorithm design in swarm robotics,” Swarm Intelligence, vol. 2, no. 2-4, pp. 209–239, 2008.

---

> ### Author Response · Authors · 2024-11-25
> **Author Response**
>
> Thanks for the feedback. As noted by the other two reviewers, we believe our presentation is clear. It is unclear to us why the reviewer stated that our paper lacks an introduction section, background information, and an approach section. Our submission includes all of these elements. Specifically, we have a preliminary section that provides background information. Section ControlAgent outlines our approach. One minor typo we have in the submission is that we mistakenly named the "introduction" section as "instruction". However,  it is quite obvious that the first section in our paper is an introduction to our paper.
>
> That being said, we have made additional efforts to further improve the paper and address potential concerns. Specifically, we have:
>
> 1. Expanded Section B to include more detailed background information.
> 2. Added real-life application scenarios to Section C to better illustrate the practical relevance of our framework.
> 3. Included additional experimental details in Section D to further support our methodology.
>
> We hope these updates strengthen the paper and address the reviewer’s concerns.

---

### Official Review · Reviewer_AbjT · 2024-10-31

**Soundness:** 2
**Presentation:** 4
**Contribution:** 3
**Rating:** 6
**Confidence:** 4

**Summary:**

This paper introduces ControlAgent, a framework that automates control system design by integrating large language model (LLM) agents with domain expertise. The framework uses multiple collaborative agents to emulate human iterative design processes, gradually tuning controller parameters to meet user-specified requirements for stability, performance, and robustness. ControlAgent consists of a central agent that analyzes tasks and distributes them to specialized agents, task-specific agents that handle detailed controller design for different system types, a Python computation agent that performs control calculations and evaluations, and a history and feedback module that enables iterative refinement of designs. The system addresses the inherent complexity of control design by breaking down the process into manageable steps and incorporating domain knowledge into the decision-making process. The authors also develop ControlEval, an evaluation benchmark comprising 500 control tasks across various system types including first-order, second-order, systems with delay, and higher-order systems, with different response modes and specific performance criteria. This benchmark serves as a standardized way to evaluate control design workflows.

**Strengths:**

The core strength of this paper lies in how it successfully addresses the fundamental performance-robustness trade-offs inherent in classical control theory. The framework intelligently uses loop-shaping and PID tuning methodologies, employing settling time and phase margin as key tuning parameters - a sophisticated approach that mirrors established control engineering practices. The iterative design process is noteworthy for its theoretical soundness. Rather than treating controller design as a single-shot optimization problem, ControlAgent mimics the systematic approach used by human experts, progressively refining controller parameters while managing the complex interplay between performance metrics. The empirical results validate this approach, showing success across various system types and complexity levels, with particularly impressive results in handling unstable and higher-order systems. The framework's ability to achieve 100% success rates for first-order and stable second-order systems, while maintaining high performance even for complex higher-order and unstable systems, demonstrates its robust theoretical foundation and practical effectiveness.

**Weaknesses:**

- The evaluation methodology raises several concerns. While ControlEval includes 500 control tasks, the paper doesn't clearly justify the distribution of these tasks or demonstrate their representativeness of real-world control problems. The generation process for higher-order systems is particularly problematic - the authors admit to manually designing these cases, which could introduce bias and may not reflect the true complexity of higher-order system control.
- The comparison with baselines is somewhat limited. The paper primarily compares against relatively simple LLM-based approaches (zero-shot, few-shot) and a single traditional tool (PIDtune). Modern control design often employs more complex methods like robust control, model predictive control, or optimization-based approaches, which are notably absent from the comparison. The performance metrics are also relatively basic, focusing mainly on settling time and phase margin while overlooking other important characteristics like disturbance rejection and noise sensitivity.
- The iterative design process lacks theoretical guarantees of convergence or optimality. The paper doesn't provide analysis of when or why the iteration process might fail, nor does it establish bounds on the number of iterations needed for convergence.
- The framework's heavy reliance on proprietary LLM models raises questions about reproducibility and practical deployment. The authors don't thoroughly explore how the system's performance might vary with different base LLMs or how it might degrade with smaller, more practical models.

**Questions:**

- How does ControlAgent handle model uncertainty? While you discuss robustness through phase margin, could you elaborate on whether the framework considers parametric uncertainties or unmodeled dynamics?
- For higher-order systems, you mention manual design of 50 cases. Could you explain your methodology for ensuring these cases are representative and unbiased? What criteria guided your selection?
- For the history and feedback module, how do you handle the context window limitations of LLMs? Could you provide more details about the memory management strategy?
- Could you provide a more detailed analysis of failure cases, particularly for higher-order systems where performance was lower? Understanding these cases would help assess the framework's limitations.

---

> ### Author Response · Authors · 2024-11-25
> **Author Response, Part 1**
>
> We sincerely thank the reviewer for the constructive feedback and have provided detailed responses below to address all the reviewer’s comments carefully. Based on the feedback, we have made several significant updates to the paper: clarified higher-order dataset generation process, expanded ControlEval with new results on 10 real-world control application problems, demonstrated ControlAgent on a real DC motor in the physical real world, evaluated the performance of ControlAgent using a more accessible and smaller LLM (Llama-3.1-70B), and provided additional robustness evaluation results for controllers designed by ControlAgent. We hope these updates address the reviewers' concerns and invite you to reevaluate our paper based on these new results. We welcome any further feedback and suggestions for improvement.
>
> **Q1.  The paper doesn't clearly justify the distribution of these tasks or demonstrate their representativeness of real-world control problems. The generation process for higher-order systems is particularly problematic – the authors admit to manually designing these cases, which could introduce bias and may not reflect the true complexity of higher-order system control. What criteria guided your selection?**
>
> A1: Thank you for this insightful comment. For first-order and second-order systems, we randomize important system parameters such as dominant pole, DC gain, natural frequency, and damping ratio to ensure the representativeness of the selected problems. For higher-order systems, it is true that we manually designed 50 stable and unstable higher-order systems along with their associated control requirements for ControlEval. For representativeness, we manually add two or three extra poles to 1st/2nd order systems. We make sure that the problems cover both the case where the added poles are far away from the original 1st/2nd-order poles and the case where the added poles are close to the original poles. We use our own control domain expertise to ensure that the resulting systems remained controllable and adhered to practical design requirements, such as there existing a controller for the designed system to achieve desired settling time and phase margin. A subtle point that should be emphasized is that the generation of higher-order systems cannot be completely random since there are higher-order systems that are fundamentally difficult to control (e.g., the system with the right half plane zero and the right half pole being close to each other is fundamentally difficult to control) and such systems are avoided by practical system design so that we should not include them in our evaluation set. Therefore, it is quite difficult to have a fully randomized generation process for higher-order systems. We definitely agree with the reviewer that we need to improve the representativeness of real-world control problems. To achieve this, we have added two new sets of important results: i) we expanded ControlEval with a new application category consisting of 10 real-world control design application problems from diverse resources, ii) we evaluated ControlAgent and all the baseline methods on a real DC motor control task in the physical real world. More discussions are given in our next response (Part 2).

---

> ### Author Response · Authors · 2024-11-25
> **Author Response, Part 2**
>
> **Representativeness of Real-World Control Applications**
>
> As mentioned above, to improve the representativeness of ControlEval on real-world control application, we have added two new sets of important results to our revised paper. Now details are given below.
>
> First, we expanded ControlEval with a new application category consisting of 10 real-world control design application problems from diverse resources. In addition, for these 10 real-world physical systems, we randomly generated the design requirements for settling time (for real-world applications, only upper bounds are needed to ensure fast tracking) and phase margin within proper range to reduce human bias. The information of these application problems are briefly given below.
>
> | Real-life application                | Dynamical System                                    | Settling Time       | Phase Margin        |
> |--------------------------------------|----------------------------------------------------|-----------------------------------|---------------------------------|
> | Laser Printer Positioning System     | $\frac{4(s + 50)}{s^2 + 30s + 200}$               | $T_s \le 0.36$       | $\phi_m \ge 74.24^\circ$       |
> | Space Station Orientation Control    | $\frac{20}{s^2 + 20s + 100}$                      | $T_s \le 0.64$        | $\phi_m \ge 76.22^\circ$       |
> | Vehicle Steering Control System      | $\frac{1}{s(s+12)}$                               | $T_s \le 0.58$        | $\phi_m \ge 56.98^\circ$       |
> | Antenna Azimuth Control System       | $\frac{20.83}{s^2 + 101.7s + 171}$               | $T_s \le 1.57$        | $\phi_m \ge 82.95^\circ$       |
> | Autonomous Submersible Control       | $\frac{-0.13(s+0.44)}{s^2 + 0.23s + 0.02}$        | $T_s \le 41.49$      | $\phi_m \ge 69.49^\circ$       |
> | Aircraft Pitch Control System        | $\frac{1.151s+0.1774}{s^3+0.739s^2+0.92s+1}$      | $T_s \le 33.58$      | $\phi_m \ge 53.92^\circ$       |
> | Missile Yaw Control System           | $\frac{-0.5(s^2+2500)}{(s-3)(s^2+50s+1000)}$      | $T_s \le 3.95$        | $\phi_m \ge 63.43^\circ$       |
> | Helicopter Pitch Control System      | $\frac{25(s+0.03)}{(s+0.4)(s^2-0.36s+0.16)}$      | $T_s \le 30.36$      | $\phi_m \ge 66.81^\circ$       |
> | Speed Control of a Hard Disk Drive   | $\frac{-0.1808s^4-0.5585s^3+0.4249s^2-8.625s+135.1}{s^4+0.2046s^3+8.932s^2+0.1148s+0.007285}$ | $T_s \le 88.11$      | $\phi_m \ge 52.22^\circ$       |
> | High-Speed Train Control System      | $\frac{12}{(s+10)(s+70)}$                         | $T_s \le 2.08$       | $\phi_m \ge 74.40^\circ$       |
>
> The details of these applications and related resources are provided in Appendix C.1 of our revised paper.  We have also obtained the evaluation results for ControlAgent and baseline methods as below:
>
> | Method            | Zero-shot | Zero-shot CoT | Few-shot | Few-shot CoT | PIDTune | ControlAgent |
> |--------------------|-----------|---------------|----------|--------------|---------|--------------|
> | Success Rate (%)   | 10        | 0             | 20        | 0            | 50      | 100          |
>
> One can clearly see that ControlAgent outperforms all the baselines in this new class of real-world control application problems. More details can be found in Appendix C.1 of our revised paper.
>
> **Implement ControlAgent with a real-world DC motor control task.**
>
> Secondly, we evaluated ControlAgent and all the baseline methods on **a real-world DC motor** control task, see Figure 7 in the revised paper for demonstrations. The model used for design is a third-order model in the form of
> $T(s)=\frac{K }{s\left((L_a s+R_a)(J s+b)+K_\tau K_v \right)},$  where there is a gap between the model and the real motor that the control is deployed. Specifically, we implement the designed controllers from ControlAgent and all the baseline methods on a real DC motor and provide real-world data on the controller performances. The results in Figure 8 again demonstrate that ControlAgent gives much better design than all the baseline methods when the designed controllers are deployed on a real system (details are given in Appendix C.2 of our revised paper).
>
> We believe that our new results have convincingly demonstrated the practical use of ControlAgent across a wide range of real-world control tasks, significantly improving the empirical soundness of our work.

---

> ### Author Response · Authors · 2024-11-25
> **Author Response, Part 3**
>
> **Q2 (Part 1). The comparison with baselines is somewhat limited. The paper primarily compares against relatively simple LLM-based approaches (zero-shot, few-shot) and a single traditional tool (PIDtune). Modern control design often employs more complex methods like robust control, model predictive control, or optimization-based approaches, which are notably absent from the comparison.**
>
> We agree with the reviewer that modern control design often involves advanced methodologies such as robust control (H-infinity and mu-synthesis) and MPC. However, our current focus is on classic control on PID and loop-shaping, which represents the majority of control design in industrial applications (there is a well-accepted folklore that 90% of controllers in the control industry are PID or loop-shaping). For such classic control that dominates the control industry, PID tuning methods would be the most direct competitor and we have compared ControlAgent against PIDTune, which is popular for industry use.
>
> Our work also motivates the need of future study on developing LLM agents to design H-infinity robust controllers and MPC schemes with the right cost function and time horizon for the remaining 10% of advanced control tasks in industry. For such future study, it makes sense to compare to existing robust control or MPC methods that require human experts for setting up (for example, traditionally, human experts need to carefully set up the so-called "weighting functions" such that robust control can be applied to a specific problem).  We emphasize that the use of robust control methods or MPC methods even require more human expertise and computational resources than PID tuning, and this is the main reason why they are much less popular than classic PID/loop-shaping control methods for control industry.
>
> **Q2 (Part 2). The performance metrics are also relatively basic, focusing mainly on settling time and phase margin while overlooking other important characteristics like disturbance rejection and noise sensitivity.**
>
> Phase margin, settling time, and stability are fundamental metrics that are widely used in control industry. These metrics are well-defined over a wide class of problems and in general less problem-dependent, capturing the performance/robustness trade-off in classic control design. In contrast, disturbance rejection and noise sensitivity are more problem-dependent in the sense that disturbance and sensor noise can have very different patterns for different application tasks (IID Gaussian assumptions are not widely used in control industry, and the evaluations on disturbance and noise rejection eventually require hardware experiments). Given the current resources we have, we added the study on noise sensitivity for a real DC motor in the physical real world. We show that the controller designed by ControlAgent achieves good reference tracking and noise rejection simultaneously on a real DC motor under real sensor noise. Please see our discussions in Appendix C.1 of our revised paper.
>
> **Q3. The iterative design process lacks theoretical guarantees of convergence or optimality. The paper doesn't provide analysis of when or why the iteration process might fail, nor does it establish bounds on the number of iterations needed for convergence.**
>
>
> Thank you for raising this important concern about the theoretical guarantees of convergence and optimality in the iterative design process. We want to argue that even human control engineers cannot guarantee that they can always converge to a successful PID control design. Since ControlAgent mimics how human practicing controllers to design control systems, it seems natural to expect that ControlAgent will not have theoretical convergence guarantees. We agree with the reviewer that we should provide analysis for when or why the iteration process of ControlAgent might fail. We provide such failure analysis in Section D.4.5 of our revised paper. Please also see our response to **Q7** below in (Author Response, Part 5).

---

> ### Author Response · Authors · 2024-11-25
> **Author Response, Part 4**
>
> **Q4. The framework's heavy reliance on proprietary LLM models raises questions about reproducibility and practical deployment. The authors don't thoroughly explore how the system's performance might vary with different base LLMs or how it might degrade with smaller, more practical models.**
>
> We thank the reviewer for the thoughtful comments regarding the reliance on proprietary LLM models and concerns about reproducibility and practical deployment. To address this, we have included additional experiments using a more accessible, open-source LLM backbone: Llama-3.1-70b.
>
> We evaluated ControlAgent with the Llama model on four representative tasks: three first-order stable systems (with fast, moderate, and slow response modes) and a more challenging task involving higher-order systems. The results are provided below, where we report the metrics ASR, AgSR, and pass@k with $k=1,3,5$ (the formal definition of pass@k metric can be found in Section D.2). Additionally, we include the averaged number of samples required, stability margins designed by ControlAgent for completeness.
>
> |            | 1st fast | 1st moderate | 1st slow | higher  |
> |------------|----------|--------------|----------|---------|
> | pass@1     | 0.927    | 1.000        | 0.996    | 0.300   |
> | pass@3     | 1.000    | 1.000        | 1.000    | 0.446   |
> | pass@5     | 1.000    | 1.000        | 1.000    | 0.480   |
> | # samples  | 3.053    | 2.016        | 2.824    | 24.500  |
>
> It can be seen that ControlAgent with Llama-3.1-70b is also effective for simpler, first-order control tasks but faces challenges with more complex, higher-order systems.  The $pass@1$ rate is only 0.300, indicating that the model struggles to solve the problem on the first attempt. The $pass@3$ and $pass@5$ rates improve to 0.446 and 0.480, respectively, but still remain below 50\%, suggesting that the task is considerably more challenging for the model.
>
> In addition, the average number of iterations required for first-order stable systems is relatively low, with moderate response mode requiring the fewest iterations (2.016), followed by slow (2.824) and fast (3.053). In contrast, the higher-order system requires a significantly higher average number of iterations (24.5), reflecting the increased complexity and difficulty of the task.
>
> **Q5. How does ControlAgent handle model uncertainty? While you discuss robustness through phase margin, could you elaborate on whether the framework considers parametric uncertainties or unmodeled dynamics?**
>
> Thank you for your thoughtful question regarding how ControlAgent handles model uncertainty, including parametric uncertainties and unmodeled dynamics. ControlAgent heavily relies on loop-shaping, which is well-known to yield good gain margins (for parametric uncertainty), and phase margin (for phase variations and tolerance of time delays). In addition, the famous loop-shaping theorem states that loop-shaping also yield a general disk margin for a value at least at 0.4. The disk margin provides a comprehensive robustness measure by simultaneously addressing both gain margin and phase margin, providing robustness against non-parametric unmodeled dynamics. To summarize, although phase margin and settling time are used as tuning knobs for ControlAgent, reasonably good gain margins and disk margins are also automatically addressed by ControlAgent.
> We have added a new disk margin analysis of ControlAgent in Section D.4.4.
>
> The following table demonstrates the disk margins for controllers designed by ControlAgent, showing robust stability across various control problems.
>
> |             | 1st stb f         | 1st stb m         | 1st stb s     | 2nd stb f         | 2nd stb m         | 2nd stb s         | 1st un        | 2nd un        | 1 w dly          | High st       |
> |-------------|---------------|---------------|-----------|---------------|---------------|---------------|---------------|---------------|---------------|---------------|
> | Disk Margin | 1.6184 ± 0.0238 | 1.9833 ± 0.0046 | 2 ± 0     | 1.2553 ± 0.0255 | 1.2852 ± 0.0304 | 1.4498 ± 0.0134 | 1.0695 ± 0.0976 | 0.9646 ± 0.0257 | 0.6628 ± 0.2021 | 1.1058 ± 0.0693 |
> | Gain Margin | [0.1074, Inf]  | [0.0045, Inf]  | [0, Inf]  | [0.2317, 4.6367] | [0.2211, 4.9703] | [0.1609, 6.5571] | [0.3157, 4.4090] | [0.3532, 2.9645] | [0.5492, 2.3591] | [0.3014, Inf]   |
> | Phase Margin| [-77.7608, 77.7608] | [-89.4855, 89.4855] | [-90, 90]  | [-63.9961, 63.9961] | [-65.1729, 65.1729] | [-71.7560, 71.7560] | [-55.3823, 55.3823] | [-51.2628, 51.2628] | [-35.2660, 35.2660] | [-56.8894, 56.8894] |
>
>
> These results show that ControlAgent maintains adequate robustness margins even under varying conditions, reinforcing its capability to handle parametric uncertainty and unmodeled dynamics effectively.

---

> ### Author Response · Authors · 2024-11-25
> **Author Response, Part 5**
>
> **Q6. For the history and feedback module, how do you handle the context window limitations of LLMs? Could you provide more details about the memory management strategy?**
>
> Thank you for your question regarding our approach to memory management and handling context window limitations. In ControlAgent, we address these limitations by selectively storing and providing only the essential historical information to the LLMs based on our own control expertise. Specifically, we retain key data such as design parameters, performance metrics, and feedback from previous iterations, while excluding unnecessary details from the LLMs’ responses. This ensures that the memory buffer remains compact and efficient.
> For example, here is an illustration of two historical designs stored in the memory buffer for one specific task:
>
> ```
> ### Design 1
> Parameters:
> - omega_L = 5.5
> - beta_b = 3.162
> - beta_l = 3.162
>
> Performance:
> - phase_margin = 15.89
> - settling_time = 3.20
> - steadystate_error = 0.0
>
> Feedback:
> - Phase margin should be at least 52.82 degrees.
>
> ### Design 2
> Parameters:
> - omega_L = 6.5
> - beta_b = 4.0
> - beta_l = 4.0
>
> Performance:
> - phase_margin = 30.06
> - settling_time_min = 3.60
> - steadystate_error = 0.0
>
> Feedback:
> - Phase margin should be at least 52.82 degrees.
> ```
>
> Only the above summarized history is fed into the LLM for the next iteration. This strategy allows the LLM to focus on refining the design based on the key feedback and performance metrics, without exceeding the context window limitations.
> From our observations, this approach effectively prevents context window overflow while maintaining the iterative design process to be effective. Additionally, it ensures memory efficiency by retaining only the critical information required to improve upon previous designs. We have added the above discussions in Section D.3.
>
> **Q7. Could you provide a more detailed analysis of failure cases, particularly for higher-order systems where performance was lower? Understanding these cases would help assess the framework's limitations.**
>
> Thank you for this thoughtful question. We have added a new failure mode analysis of ControlAgent for higher-order system design in Section D.4.5. Specifically, we identified several failure modes in the LLM's approach to controller design, each revealing challenges in reasoning and parameter adjustment strategies:
> 1. **Calculation Errors**: One notable failure occurred with a marginally unstable system featuring a double integrator. The LLM incorrectly calculated the minimum loop bandwidth as below: "*The fastest unstable pole is at 0, so we initially chose $\omega_L = 2.5 \times 0 = 2.5.$*" This calculation was incorrect and did not align with proper design principles.
>
>
> 2. **Incomplete Parameter Adjustments**: The LLM often adjusted only two parameters ($\omega_L$ and $\beta_l$), neglecting $\beta_b$, which is crucial for balancing the settling time and phase margin. For example, in one design, the final parameters were $\omega_L = 60$, $\beta_b = 0.8$, and $\beta_l = 1000$, with $\beta_b$ remaining unchanged throughout iterations. This limited adjustment scope hindered optimal design.
>
>
> 3. **Hallucination Errors**: Another failure involved misidentifying the dominant pole. In one instance, the LLM incorrectly identified -50 as the dominant pole instead of the actual dominant poles at $-2.1 \pm 2.142$:
>  "*The poles at -50 and $-2.1\pm2.14242853j$ suggest a relatively fast response due to the dominant pole at -50.*"
>  This misunderstanding led to incorrect design decisions.
>
>
> These examples highlight key areas where the LLM's reasoning and parameter optimization strategies need improvement. Addressing these failure modes is a priority for future iterations of the framework.

---

> ### Author Response · Authors · 2024-12-01
> **Acknowledgement and Follow-up**
>
> Dear Reviewer,
>
>   We want to thank you again for your valuable comments and constructive feedback that helped us improve our paper significantly. In response to your suggestions, we have made significant improvements to the paper including: adding more evaluations with open-sourced model; expanding ControlEval with 10 more tasks related to real-world applications; demonstrating ControlAgent on the hardware of a real DC motor control system in the physical real world; adding more evaluations with disk margins. We believe that such revision has significantly improved our paper, and we hope that all your concerns have been addressed. With the discussion period closing soon, please feel free to let us know if you have any further questions or feedback. Thanks so much!
>
>
> Sincerely,
>
> Authors

---

> > ### Comment · Reviewer_AbjT · 2024-12-02
> > **All Concerns Addressed Successfully**
> >
> > Thank you for your response. All my concerns have been resolved.

---

> > > ### Author Response · Authors · 2024-12-03
> > > **Thank You for Your Feedback**
> > >
> > > We sincerely thank the reviewer for their thoughtful feedback and continued positive support. We are delighted that our response has successfully addressed all of your concerns. If you have any additional questions or comments, please do not hesitate to share them with us, we would be happy to discuss them further!

---

### Author Response · Authors · 2024-11-25
**General Response, Part 3**

**Relevance to the ICLR Community**

Reviewer W6Fy has asked about our paper's relevance to ICLR, and we believe our work is particularly relevant to the ICLR community. As stated by the ICLR 2025 Reviewer Guide, a paper brings value to the ICLR community when it convincingly demonstrates new, relevant, impactful knowledge. **A key academic contribution of our paper is the novel integration of two major disciplines - control engineering and large language models.** Control engineering is not merely a domain-specific application, but rather a fundamental discipline that underpins modern technology and civilization (from aerospace and robotics to power systems and autonomous vehicles) and has its own rich mathematical foundations and engineering design principles developed over decades. Our paper presents the first approach to systematically bridging these two disciplines by showing how to encode complex control engineering principles into LLM agents and combine control-oriented computation tools with LLM reasoning to enable ControlAgent to automatically navigate the fundamental performance-robustness tradeoffs in control design. This is beyond a naive integration of control and LLMs. Our key contributions include:

1. Iterative Refinement Mechanism: Mimicking control engineers' decision-making processes to navigate performance-robustness trade-offs.
2. Structured Memory Management: Allowing LLMs to learn from prior designs while respecting context limitations.
3. Multi-Agent Architecture: Enabling specialized LLM agents to collaborate effectively, ensuring technical correctness and efficiency.


Similar to how the integration of AI with fields like physics or biology has led to groundbreaking advances (e.g., AlphaFold), our work demonstrates how combining LLMs with control engineering can push both fields forward. The insights gained here are broadly applicable to other engineering disciplines, offering valuable lessons for integrating LLMs with deep engineering domain expertise.

We also align with ICLR’s mission to explore AI applications across domains, including robotics, autonomy, and planning, as stated in the Call for Papers (https://iclr.cc/Conferences/2025/CallForPapers). Previous ICLR papers, such as LLMs for operations research or geospatial tasks [1,2], affirm that engineering applications are highly relevant. Similarly, our work advances ML methodology by enabling LLMs to perform technical reasoning and design in engineering domains requiring both deep theoretical understanding and sharp engineering design intuitions.

Our empirical validation through ControlEval and the new added results on real-world applications, systematically demonstrates ControlAgent’s effectiveness compared to traditional tools and baseline LLM approaches. The strong performance highlights the value of integrating rigorous engineering principles with LLM reasoning, showcasing a practical path toward leveraging AI for real-world, complex problem-solving. We believe our work exemplifies ICLR’s tradition of advancing ML methodology through interdisciplinary innovation. While our paper focuses on control design, the broader implications for LLM-enabled engineering provide valuable insights that extend beyond a single discipline.

[1] Xiao, Z., Zhang, D., Wu, Y., Xu, L., Wang, Y.J., Han, X., Fu, X., Zhong, T., Zeng, J., Song, M. and Chen, G., 2023. Chain-of-Experts: When LLMs Meet Complex Operations Research Problems. ICLR.

[2] Manvi, R., Khanna, S., Mai, G., Burke, M., Lobell, D. and Ermon, S., 2023. Geollm: Extracting geospatial knowledge from large language models. ICLR.

---

### Author Response · Authors · 2024-11-25
**General Response, Part 2**

**New Results in Revision for Improving Empirical Soundness**

Reviewer  AbjT gives a really valuable comment on that we need to demonstrate the representativeness of ControlEval for real-world control problems. To address this comment, we have obtained two new important results.

First, we expanded ControlEval with a new application category consisting of 10 real-world applications from diverse resources. In addition, for these 10 real-world physical systems, we randomly generated the design requirements for settling time and phase margin within proper range to reduce human bias. For this category, we just sample the upper bounds of settling time since for real-life applications it is important ensure the fast tracking. The information of these application problems are briefly given below.

| Real-life application                | Dynamical System                                    | Settling Time       | Phase Margin        |
|--------------------------------------|----------------------------------------------------|-----------------------------------|---------------------------------|
| Laser Printer Positioning System     | $\frac{4(s + 50)}{s^2 + 30s + 200}$               | $T_s \le 0.36$       | $\phi_m \ge 74.24^\circ$       |
| Space Station Orientation Control    | $\frac{20}{s^2 + 20s + 100}$                      | $T_s \le 0.64$        | $\phi_m \ge 76.22^\circ$       |
| Vehicle Steering Control System      | $\frac{1}{s(s+12)}$                               | $T_s \le 0.58$        | $\phi_m \ge 56.98^\circ$       |
| Antenna Azimuth Control System       | $\frac{20.83}{s^2 + 101.7s + 171}$               | $T_s \le 1.57$        | $\phi_m \ge 82.95^\circ$       |
| Autonomous Submersible Control       | $\frac{-0.13(s+0.44)}{s^2 + 0.23s + 0.02}$        | $T_s \le 41.49$     | $\phi_m \ge 69.49^\circ$       |
| Aircraft Pitch Control System        | $\frac{1.151s+0.1774}{s^3+0.739s^2+0.92s+1}$      | $T_s \le 33.58$     | $\phi_m \ge 53.92^\circ$       |
| Missile Yaw Control System           | $\frac{-0.5(s^2+2500)}{(s-3)(s^2+50s+1000)}$      | $T_s \le 3.95$       | $\phi_m \ge 63.43^\circ$       |
| Helicopter Pitch Control System      | $\frac{25(s+0.03)}{(s+0.4)(s^2-0.36s+0.16)}$      | $T_s \le 30.36$     | $\phi_m \ge 66.81^\circ$       |
| Speed Control of a Hard Disk Drive   | $\frac{-0.1808s^4-0.5585s^3+0.4249s^2-8.625s+135.1}{s^4+0.2046s^3+8.932s^2+0.1148s+0.007285}$ | $T_s \le 88.11$     | $\phi_m \ge 52.22^\circ$       |
| High-Speed Train Control System      | $\frac{12}{(s+10)(s+70)}$                         | $T_s \le 2.08$      | $\phi_m \ge 74.40^\circ$       |

 We have obtained the evaluation results for ControlAgent and baseline methods as below:

| Method            | Zero-shot | Zero-shot CoT | Few-shot | Few-shot CoT | PIDTune | ControlAgent |
|--------------------|-----------|---------------|----------|--------------|---------|--------------|
| Success Rate (%)   | 10        | 0             | 20        | 0            | 50      | 100          |

One can clearly see that ControlAgent outperforms all the baselines in this new class of real-world control application problems. More discussions will be provided in the individual response to Reviewer AbjT and included in Appendix C.1 of our revised paper.

Secondly, we evaluated ControlAgent and all the baseline methods on a real DC motor control task in the physical real world (the model used for design is a third-order model in the form of $T(s)=\frac{K }{s\left((L_a s+R_a)(J s+b)+K_\tau K_v \right)}$ but there is a gap between the model and the real motor that the control is deployed). Specifically, we implement the designed controllers from ControlAgent and all the baseline methods on a real DC motor and provide real-world data on the controller performances. The results again demonstrate that ControlAgent gives much better design than all the baseline methods when the designed controllers are deployed on a real system (details will be given in Appendix C.2 of our revised paper).

We believe that our new results have convincingly demonstrated the practical use of ControlAgent across a wide range of real-world control tasks, significantly improving the empirical soundness of our work.

---

### Author Response · Authors · 2024-11-25
**General Response, Part 1**

We sincerely thank all the reviewers for the feedback and comments. We start with a common response clarifying the theoretical/empirical soundness of our approach and the relevance of our paper to the ICLR community. We will address all the review comments in a more detailed manner in the individual responses provided for each reviewer.

**Theoretical Soundness of Our Paper**

As commented by Reviewer  AbjT, the iterative design process of ControlAgent is noteworthy for its theoretical soundness. Our ControlAgent framework is the first LLM agent framework in automating the process of designing controllers that can rigorously meet pre-specified requirements on performance (settling time) and robustness (phase margin) and effectively navigate the fundamental performance/robustness trade-off in classic control design. Reviewer W6Fy is skeptical on whether the overall ControlAgent system can work without a human in the loop or method for filtering correct answers. Here we emphasize that ControlAgent is fully automated and there is no human in the loop. Specifically, ControlAgent uses a Python computation sub-agent to automatically and exactly evaluate the stability, settling time, and phase margin of the designed controllers, and put such exact information into a memory module such that the LLM sub-agents exactly know whether the current designed controller meets the user-specified performance/robustness requirements or not. The Python computation sub-agent is integrated as a module of ControlAgent and works in a fully automated manner to ensure the ControlAgent always know the true settling time and phase margin of its own designed controllers. To summarize, our approach is technically sound in: i) LLM sub-agents are set up to automatically mimic how practicing engineers reason about the tuning of PID or loop-shaping controllers, ii) Python computation sub-agents and the memory module are integrated to ensure that ControlAgent can provide performance/robustness guarantees for its own control design in a fully automated manner.

---

### Meta-Review · Area_Chair_b4aW · 2024-12-20

**Metareview:**

This paper's core strength is its demonstration that LLMs can be used to design controllers for a wide variety of different systems with specific stability, phase margin, and settling times. The system, ControlAgent, runs in a loop where the LLM designed controller is run and the system provides feedback based on the performance of a set of designs. The paper convincingly demonstrates that LLMs can be used for this purpose, and ControlAgent excels across many different control settings with high success rates even when the dynamics are complex and unstable. Designing controllers with machine learning models is a highly relevant application area to ICLR, and knowing that LLMs can do this effectively is a valuable contribution.

However, the main weakness, although not pointed out by reviewers, is the lack of contextualization of ControlAgent in the exceptionally large literature on using machine learning for control problems. There is no section in the related work that addresses the very well researched area on machine learning control (see https://faculty.washington.edu/sbrunton/mlcbook/CH02_MLC.pdf for an introduction). There is similarly no comparison to a baseline which uses a machine learning model *other* than an LLM for controller design.

Without grounding in this literature, the relevant sub-community at ICLR will not engage with this paper. There is similarly no discussion of what the benefits of an LLM over an alternative ML model are for controller design, which would be very valuable for making the paper more impactful. As a result, this paper cannot be accepted in its current form. Fixing the contextualization with respect to the ML for control community for a future version of the paper will make the paper significantly stronger.

**Additional Comments On Reviewer Discussion:**

The authors did a commendable job at answering reviewer questions. Most significantly, reviewer AbjT brought up concerns about the evaluation of the approach, and the authors provided new results to address this concern during the rebuttal period. Furthermore, the authors ran one of the designed controllers on a real DC motor, with positive results. This is an impressive feat to have accomplished during the rebuttal period.

One reviewer also raised concerns about the evaluation methodology, and the authors presented an alternative (using pass@k) in the rebuttal, which further underscores the strength of the LLM for controller design.

The authors also added sections to the Appendix to discuss guarantees that are possible using ControlAgent.

Reviewer tosD raised concerns about the scientific writing of the paper, but in my reading of the paper, the scientific writing was adequate, and their review was therefore ignored in making the final decision on this paper.

---

### Decision · Program_Chairs · 2025-01-22

Reject